# Stable isotope signatures of soil nitrogen on an environmental-geomorphic gradient within the Congo Basin

Simon Baumgartner[1,2], Marijn Bauters[2,3], Matti Barthel[4], Travis William Drake[4], Landry Cizungu Ntaboba[5], Basile Mujinya Bazirake[6], Johan Six[4], Pascal Boeckx[2], Kristof Van Oost[1]

[1]Earth and Life Institute, UCLouvain, Louvain-la-Neuve, 1348, Belgium
[2]Department of Green Chemistry and Technology, Ghent University, Ghent, 9000, Belgium
[3]Department of Environment, Ghent University, Ghent, 9000, Belgium
[4]Department of Environmental Systems Science, Swiss Federal Institute of Technology, ETH Zurich, Zurich, 8092, Switzerland
[5]Départment d'Agronomie, Université Catholique de Bukavu, Bukavu, DR Congo
[6]Department of General Agricultural Sciences, University of Lubumbashi, DR Congo

*Correspondence to*: Simon Baumgartner (simon.baumgartner@uclouvain.be)

**Abstract.**

Nitrogen (N) availability can be highly variable in tropical forests on regional and local scales. While environmental gradients influence N cycling on a regional scale, topography is known to affect N availability on a local scale. We compared natural abundance of $^{15}$N isotopes of soil profiles in tropical lowland forest, tropical montane forest, and subtropical Miombo woodland within the Congo Basin as a proxy to assess ecosystem-level differences in N cycling. Soil $\delta^{15}$N profiles indicated that N cycling in the montane forest is relatively more closed and dominated by organic N turnover, whereas the lowland forest and Miombo woodland experienced a more open N cycle dominated by inorganic N. Furthermore, we examined the effect of slope gradient on soil $\delta^{15}$N within forest types to quantify local differences induced by topography. Our results show that slope gradient only affects the soil $\delta^{15}$N in the Miombo forest, which is prone to erosion due to a lower vegetation cover and intense rainfall at the onset of the wet season. Lowland forest, on the other hand, with a flat topography and protective vegetation cover, showed no influence of topography on soil $\delta^{15}$N in our study site. Despite the steep topography, slope angles do not affect soil $\delta^{15}$N in the montane forest, although stable isotope signatures exhibited higher variability within this ecosystem. A pan-tropical analysis of soil $\delta^{15}$N values (i.e. from our study and literature) reveals that soil $\delta^{15}$N in tropical forests is best explained by factors controlling erosion, namely mean annual precipitation, leaf area index, and slope gradient. Erosive forces vary immensely between different tropical forest ecosystems and our results highlight the need of more spatial coverage of N-cycling studies in tropical forests, to further elucidate the local impact of topography on N cycling in this biome.

## 1 Introduction

The nutrient status of forests is an important determinant for the allocation of sequestered carbon (C) to biomass (Vicca et al., 2012), with nitrogen (N) and/or phosphorus (P) limitation potentially constraining net primary productivity (NPP) (Alvarez-

Clare et al., 2013; Fernández-Martínez et al., 2014; Townsend et al., 2011; Vitousek et al., 2010). In addition, models show that the increase of primary production due to $CO_2$ fertilization will be constrained by nutrient limitations (Wieder et al., 2015). Therefore, understanding nutrient cycling in forests globally is key to assess present and future forest productivity (Townsend et al., 2011; Wieder et al., 2011). Given their major role in the global C cycle (Lewis, 2006; Malhi and Grace, 2000), quantifying nutrient supply and availability is especially important for tropical forests.

To date, research has revealed a high diversity of nutrient cycles with substantial variation from regional to local scales throughout the tropics. Old and highly weathered soils in tropical lowland forests are typically considered to be rich in N and poor in rock-derived P (e.g. Vitousek, 1984). In contrast, tropical montane forests that lie on steep terrain are subjected to geomorphic processes that rejuvenate soils with bedrock nutrients through uplift and erosion. In regions with a moderate uplift, P depletion is unlikely to occur due to this rejuvenation (Porder et al., 2007). As a result of reduced N supply in higher altitudes, due to slower N mineralization processes at lower temperatures, it is more likely that plant growth is limited by N availability in tropical montane forests (Bauters et al., 2017; Soethe et al., 2008; Tanner et al., 1998). Thus it is important that nutrient availability in tropical forests is not generalized across these different ecosystems. Therefore, analyses of N cycling within ecological gradients are necessary. The "openness of the N cycle" is a good metric to describe cycling differences across ecosystems with contrasting N availabilities. While in N limited ecosystems (i.e. closed N cycle) the internal cycling and recycling rates of N are relatively more important than the absolute in- and outputs (inputs: $N_2$-fixation and N deposition; outputs: gaseous and leaching losses), the opposite is true for ecosystems that are rich in N (i.e. open N cycle) (Boeckx et al., 2005).

Furthermore, at a local scale, topography has been identified as a main determinant for both biogeochemical and ecological variability in tropical forests (Asner et al., 2015; Hofhansl et al., 2020) in part through its effect on geomorphic processes. In an intact forest, slope gradient is the main control on physical erosion intensity at the hillslope scale (Roering et al., 2001). Accumulated topsoil N has shown to be affected by physical erosion (Amundson et al., 2003; Hilton et al., 2013; Perakis et al., 2015; Weintraub et al., 2015), resulting in a close relation between nutrient availability and geomorphic process rates. Therefore, while at regional scales mainly climatic and environmental gradients are responsible for differences in nutrient availabilities (Prescott, 2002; Read, 1991), hydro-geomorphic gradients seem to be a main determinant of N cycling at the local scale (Amundson et al., 2003). Some studies from geomorphic active regions of the tropics (e.g. Taiwan and Costa Rica) found lower N availability in steeper sloping positions suggesting that erosion substantially affects local N balances (Hilton et al., 2013; Weintraub et al., 2015). However, the magnitude of this effect in more stable landscapes is unknown and calls for a consistent study across geomorphic gradients in the tropics.

The natural abundance of the stable $^{15}$N isotopes ($\delta^{15}$N) of plant and soil pools is a useful proxy to assess N cycling (Amundson et al., 2003; Martinelli et al., 1999). Craine et al. (2015a) synthesized global soil $\delta^{15}$N data to understand patterns of N cycling

and showed how soil $\delta^{15}N$ values give insight into input and output pathways of N in the system (Högberg, 1990; Högberg and Johannisson, 1993). The isotopic composition of soil N is influenced by a multitude of processes and therefore interpretation of soil $\delta^{15}N$ values is challenging. The main processes controlling soil $\delta^{15}N$ values are nitrification, denitrification N mineralization, $N_2$-fixation, and N leaching. Nitrification has a large isotope fractionation effect, with fractionation factors

70    ranging from 15 to 35 ‰ (Blackmer and Bremner, 1977; Högberg, 1997). On the other hand, fractionation during denitrification is variable with fractionation factors reported ranging from 0 to 33 ‰ (Högberg 1997). Nitrification and denitrification result in a gradual enrichment of soil $\delta^{15}N$. The process of N mineralization is believed to fractionate only marginally, however, in a system with high N mineralization rates, isotopic fractionation due to mineralization can be important (Nadelhoffer and Fry 1994). $N_2$-fixation depletes soil $\delta^{15}N$ values, as it imports atmospheric N that has an isotopic value of 0

75    ‰ by default (Högberg, 1997). Aside from these processes, site- and soil-specific characteristics can control soil $\delta^{15}N$. In addition, climatic conditions such as rainfall (Austin & Vitousek, 1998; Boeckx et al., 2005), mean annual temperature (Boeckx et al., 2005) and soil moisture (Handley et al., 1999) all influence soil $\delta^{15}N$. Generally, soil $\delta^{15}N$ signals the openness of the N cycle, with more enriched values in an open system that is more prone to N losses, compared to an more closed system (with the assumption that input $\delta^{15}N$ are similar in both systems) (Boeckx et al., 2005). On a local scale, soil erosion alters the soil

80    $\delta^{15}N$ and leads to more depleted values (Hilton et al., 2013; Weintraub et al., 2015). Because soil erosion is a non-fractionating process, the combined fractionation of a system decreases when soil erosion becomes more important relative to fractionating N losses, such as denitrification. In this case, the $\delta^{15}N$ of the remaining N in the system more closely resembles the signature of the N inputs compared to systems with less erosion.

85    The objectives of this study were 1) to assess the differences in N availability and N openness in three different forest types along an environmental-geomorphic gradient in the Congo Basin and 2) evaluate the extent and 3) drivers of physical erosion and its effects on N availability in these forests. To achieve these objectives, soil N content and the stable N isotope compositions were measured in soil profiles taken from different topographical positions in three different forest ecotypes. We used soil $\delta^{15}N$ as a proxy that integrates N transformations for differences in N availability and the openness of a system. We

90    hypothesized that soil $\delta^{15}N$ values are highest in lowland tropical forest, which would be an indication for a higher N availability and openness of N cycle. Furthermore, we hypothesized that $\delta^{15}N$ of topsoil N would be lower on steeper slopes compared to $\delta^{15}N$ of topsoil N on less steep slopes and more enriched compared to deeper soil layers of the same profile, and that this effect would be more pronounced in the montane and Miombo forests, where the erosional forces are higher compared to the lowland forest.

## 2 Methods

### 2.1 Site description

We selected representative sites for the Congo Basin's major forest ecotypes (tropical montane forest, tropical lowland forest, and the semi-dry Miombo woodland). The montane forest site is situated in the Kahuzi-Biéga National Park (2.344° S, 28.746° E), northwest of the city of Bukavu in the eastern part of Democratic Republic of the Congo (DRC). Two forest subtypes are present in the Park: montane mixed forest and monodominant bamboo forest. Soil samples were taken in the mixed forests. Soils are classified as Ferralsols/Acrisols (Bauters et al., 2019) (Table 1) and was formed on basalt rock (Van Engelen et al., 2006). The topography of the sampled forest is characterized by steep slopes (Figure 1). Mean annual temperature is ~15°C and mean annual rainfall is ~1970 mm (Fick and Hijmans, 2017). Rainfall is seasonal with two peaks in April and October and a dry season that lasts from June to September.

Samples from the tropical lowland forest were taken in the Yoko Forest Reserve (0.291° N, 25.295° E), south of the city of Kisangani. The Yoko Forest Reserve consists of two forest subtypes: a mixed lowland tropical forest and a monodominant forest, where the *Gilbertiodendron dewevrei* species comprises more than 60 % of the basal area. Soil samples in the lowland forest were taken from random points within the catchment, where both forest types are present. Soils in this area are classified as deeply weathered Ferralsols (Van Ranst et al., 2010) on fluvio-aeolian deposits (Van Engelen et al., 2006). The sampling site exhibited low variation in elevation and contained mainly gentle slopes (Figure 1). Samples were taken between altitudes of 440 – 460 m a.s.l. Mean annual temperature is ~24°C and mean annual rainfall is ~1760 mm (Fick and Hijmans, 2017) (Table 1). Yoko experiences a short wet season from March to May and a longer one from August to November.

For the Miombo woodland, a site 50 km north of the city of Lubumbashi was selected (11.212° S, 27.233° E). A pristine Miombo is characterized by deciduous woodland dominated by trees of the *Brachystegia, Julbernadia* and *Isoberlinia* genera (Campbell, 1996). As charcoal production and seasonal fires are a common occurrence, soil sampling in areas with recent charcoal production activities was avoided. We classified the soils as Acrisols and Cambisols and the parent material was identified as shales (Van Engelen et al., 2006). The topography is mostly flat and the elevation ranges between 1260 and 1310 m a.s.l (Figure 1). This region experiences a wet season from October to April, with a peak rainfall in December. Mean annual rainfall is ~1250 mm and mean temperature is ~21 °C (Fick and Hijmans, 2017)(Table 1).

120

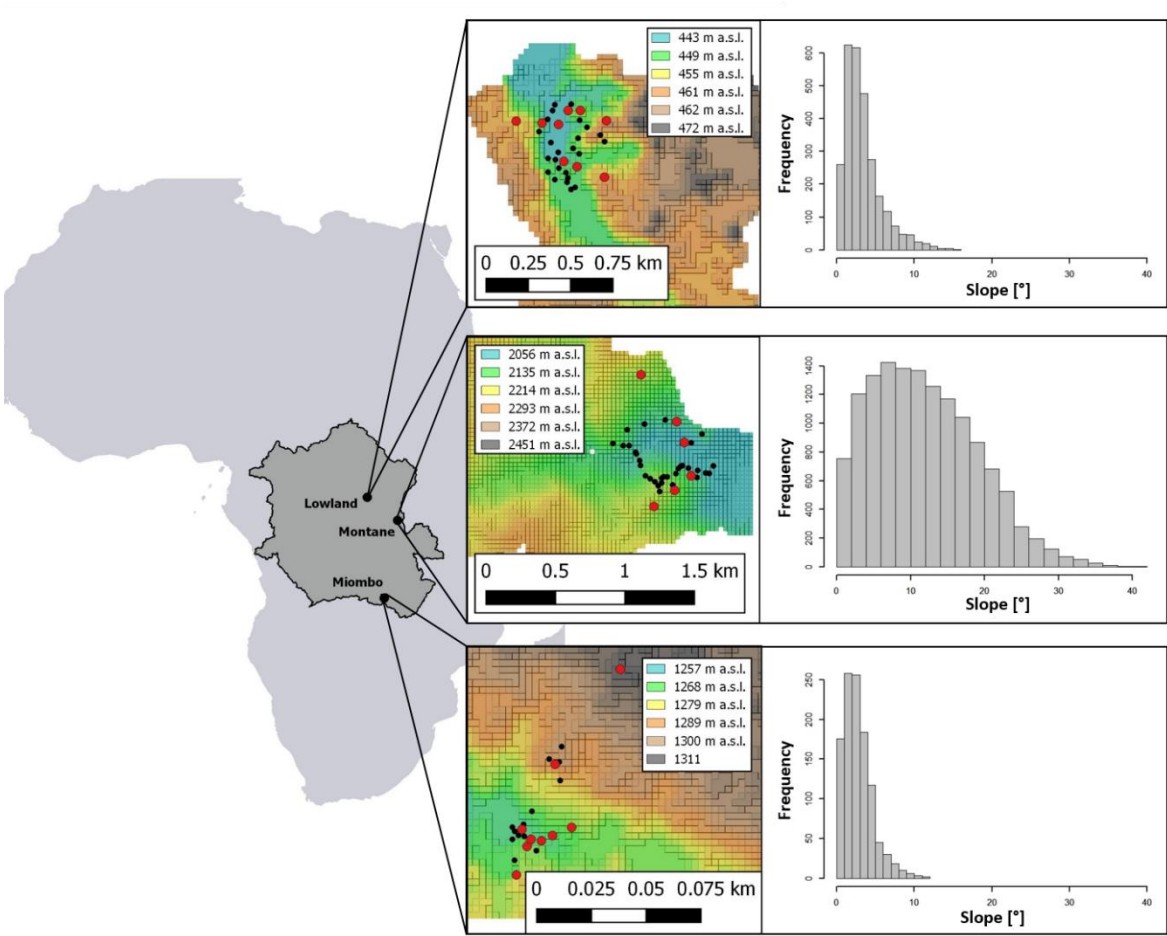

**Figure 1: Location of the sampling sites; from left to right: dark grey area indicates the Congo Basin within Africa with forest site locations, red dots indicate locations of 1m soil profiles within each forest site, black dots indicate locations of topsoil sampling positions (0-20 cm). Colors from the 30-m DEMs represent elevation (in m above sea level). Histograms on the right show the slope distribution of the respective sampling site.**

125

**Table 1: Site characteristics of the three forest types. The range of altitudes of the sampled profiles and top soil samples is shown. The slope indicates the average slope ± standard deviation from all the samples taken at the respective forest site. MAP is the mean annual precipitation and MAT is the mean annual temperature extracted from the WorldClim database (average MAP for the years 2010-2018). EC represents the mean erosion coefficient values calculated from equation 1. LSI is the landslide hazard risk for each site extracted from the Global Landslide Hazard Map (Worldbank). LAI is the mean annual leaf area index, extracted from the Copernicus LAI-300m database (European Environment Agency).**

| Forest type | Latitude [°] | Longitude [°] | Altitude [m a.s.l.] | Slope [°] | MAP [mm] | MAT [°C] | EC | LSI | Soil type | LAI |
|---|---|---|---|---|---|---|---|---|---|---|
| Montane | -2.344 | 28.746 | 2020 - 2180 | 22.8 ± 8. | 1970 | 15 | 2.79 | 3.96 | Ferralsols/Acrisols | 5.42 |
| Lowland | 0.291 | 25.295 | 440 - 460 | 4.3 ± 2.3 | 1760 | 24 | 1.22 | 1 | Ferrasols | 4.15 |
| Miombo | -11.212 | 27.233 | 1260 - 1310 | 3.9 ± 2.2 | 1250 | 21 | 20.11 | 1 | Arcrisols/Cambisols | 0.83 |

## 2.2 Sampling

To analyze differences in soil N cycling between forest sites, soil samples were taken from soil profiles at different slope positions in the catchments (slope, shoulder, and ridge) in order to better represent the catchment as a whole. Nine soil profiles were dug and sampled in both the lowland and Miombo forests. Due to difficult terrain within the montane forest, only six soil profiles were sampled (Figure 1). For each 1 m-profile, soil samples from nine different depths were taken (0-2 cm, 2-4 cm, 4-6 cm, 6-10 cm, 10-15 cm, 15-20 cm, 20-30 cm, 30-50 cm and 50-100cm). To calculate N stocks, bulk density of the soil was determined at five depths of each soil profile using a 100 cm$^3$ bulk density ring. The bulk density samples were dried at 105°C for 48h and then weighed. To analyze the effect of surface slope angles on stable isotopic signature of soil $\delta^{15}$N, additional topsoil samples (bulk sample from the top 0-20 cm) were taken from throughout the catchments at locations with varying slopes (montane: n=27; lowland: n=24; Miombo: n=21) (Figure 1). Profile- and topsoil sample positions were recorded with a GPS (GPSMAP 60CSx; Garmin; accuracy <5m) and slope angles were estimated using 30-m SRTM derived digital elevation models (DEM) (NASA JPL) which were smoothed with a low-pass filter. In addition, leaf samples from different trees were randomly collected at the lowland and montane forest site and a composite sample for each site was used to analyze foliar $\delta^{15}$N.

## 2.3 Sample analysis

Soil samples were air dried and subsequently sieved through a 2 mm sieve to remove roots and organic residues. To homogenize, soil samples were milled and analyzed for C and N content and $^{15}$N isotopes using an elemental analyzer (Automated Nitrogen Carbon Analyzer; SerCon; Crewe, UK) interfaced with an Isotope Ratios Mass Spectrometer (IRMS; 20-20, SerCon). $\delta^{15}$N laboratory standards used had a certified value of 7.81 ‰ with an uncertainty across runs of 0.07 ‰. Stable isotope ratios are reported in delta notation:

$$\delta^{15}N = \frac{R_{SMP} - R_{STD}}{R_{STD}} \qquad \text{(Eq. 1)}$$


with $R_{SMP}$ denoting the $^{15}N/^{14}N$ ratio of the sample and $R_{STD}$ that of the international reference standard AIR-$N_2$. N stocks were calculated using mass-based N content, multiplied by the bulk density of the respective depth and integrated over the whole profile depth.

## 2.4 Inter-site comparison

For a better inter-site comparison of the erosional effect we estimated a soil loss coefficient. In addition to slope, soil erosion is controlled by vegetation cover and rainfall characteristics. To incorporate these additional factors, we calculated an erosion coefficient (EC) for each surface soil sample. To calculate the EC, we used the empirical model for sediment detachment (kg $m^{-2} yr^{-1}$) from Pelletier (2012):

$$EC = S^{\frac{5}{4}}R_a e^{-L_a} \qquad \text{(Eq. 2)}$$

Where S is the tangent of the mean hillslope gradient of all samples, $R_a$ the mean annual precipitation (MAP), and $L_a$ the mean
annual leaf area index (LAI). Numerical values for LAI were extracted from the Copernicus LAI-300m satellite images produced by the Earth Observation program of the European Commission (European Environment Agency) for the year 2019. MAP data were extracted from the WorldClim database and averaged for the years 2010-2018 (Fick and Hijmans, 2017). For a pan-tropical comparison, data points from Costa Rica (Weintraub et al., 2015) and Taiwan (Hilton et al., 2013) were extracted from the supporting material of the representative studies. Only these two studies were selected for a pan-tropical
comparison, as other datasets containing soil $\delta^{15}N$ from tropical forests do not contain information on slope angles and LAI. Furthermore, in other studies, the reported coordinates are not sufficiently precise to extract slope values from global DEMs, resulting in a neglection of within site variability of slope effects.

## 2.5 Statistical analysis

To assess the effect of forest type and soil depth on $\delta^{15}N$ and mass-based C:N ratio, linear mixed effects models were fitted, controlling for topographic position via a random intercept. A third model was fitted for N-stocks. Since stocks are integrated over the whole soil profile, soil depth was omitted as an explanatory variable in the third model, including only the effect of forest type as a fixed effect, while controlling for topographic position via a random intercept. All models were fitted using maximum likelihood methods via the lmerTest package (Kuznetsova et al., 2017). To assess the relation between slope angle
and $\delta^{15}N$ of the spatial surface samples, linear regressions were applied.
Furthermore, a structural equation model (SEM) was applied to examine the relative direct and indirect importance of different variables on the soil $\delta^{15}N$ values of the five tropical forests. A SEM was chosen because it allows to include dependencies

between variables. In soil sciences, SEMs have been used for different applications, for example by implementing different pedological process information in soil mapping approaches (Angelini et al., 2017). The variables considered to explain $\delta^{15}$N

were MAP, MAT, LAI, slope, and soil C content. The SEM model was generated using the *lavaan* package version 0.6 (Rosseel, 2012) for R. All statistical analyses were conducted using the R-software (R Development Core Team, 2019).

## 3 Results

The montane forest showed the highest N stock with mean value (mean ± standard deviation (SD)) of $1.47 \pm 0.61$ t N ha$^{-1}$. N stocks were lower in the lowland forest and Miombo woodland, with $0.87 \pm 0.11$ and $0.73 \pm 0.30$ t N ha$^{-1}$, respectively (Table

2). The montane forest and Miombo woodland soils exhibited similar C:N ratios with $11.4 \pm 1.1$ in the montane and $11.2 \pm 1.8$ in the Miombo, while the C:N ratio in the lowland forest was lower with a mean value of $7.4 \pm 0.5$ (Table 2). Average bulk N and C:N values over the whole profile are reported in Figure S2. In general, soil profiles of the Miombo woodlands showed the lowest $\delta^{15}$N values ($3.59 \pm 0.73$ ‰, average for 0 - 100 cm) than the montane forest ($5.83 \pm 1.30$ ‰) and the lowland forest ($7.62 \pm 0.63$ ‰), while the lowland forest displayed higher $\delta^{15}$N values, especially in top soils (Figure 2). Lowland values

increased steadily until a depth of 10 cm and decreased from there again until 100 cm depth. $\delta^{15}$N in the Miombo woodlands showed the highest enrichment at a depth of 30 cm. $\delta^{15}$N values of the montane forest varied considerably (large confidence intervals, Figure 2), but showed a general steady enrichment with increasing depth (Figure 2). Spatial topsoil samples from each catchment (0 - 20 cm) reflected those of the soil profiles, showing highest $\delta^{15}$N values in the lowland forest (mean ± SD; $7.55 \pm 0.69$ ‰), while the montane forest displayed slightly lower values ($6.48 \pm 1.42$ ‰), the Miombo woodland was most

depleted in $\delta^{15}$N ($2.93 \pm 0.46$ ‰) (Figure 2b). The topsoil samples from the montane forest showed highest variability in $\delta^{15}$N values, ranging from 3.71 – 10.72 ‰ with a CV of 22 %, whereas the Miombo showed a CV of 16 % and the lowland forest a CV of 9 %. Slope angle did not have a significant effect on topsoil $\delta^{15}$N in the lowland and montane forests (p=0.8 for lowland and p=0.57 for montane forest, respectively; Figure 3). In contrast, topsoil $\delta^{15}$N significantly decreased with increasing sloping angles in the Miombo ($r^2$=0.30, Figure 3). The Miombo woodland displayed the highest average EC (20.11), whereas

the montane (2.77) and lowland (1.22) forests showed clearly lower values (Table 1).

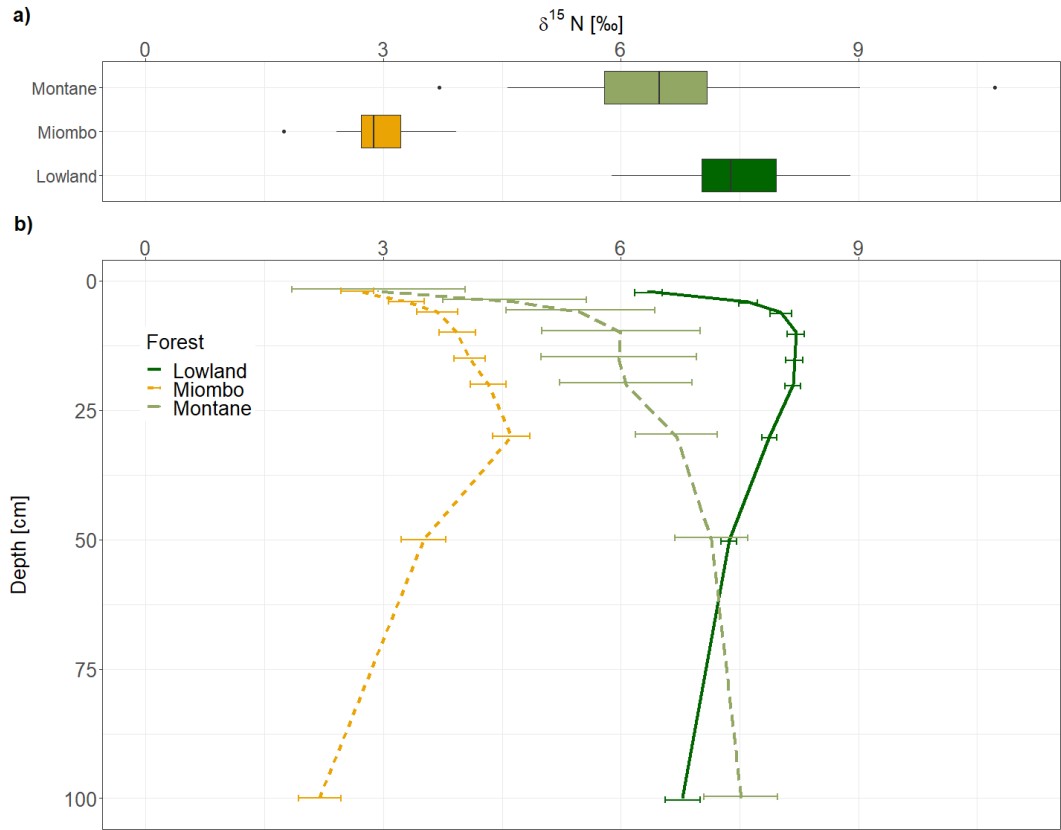

**Figure 2: a) Boxplots of the δ¹⁵N of the topsoil (0-20 cm) samples from the montane, lowland and Miombo forest sites. b) δ¹⁵N profiles from the different forest types. Lines showing the means of the replicates from one sampling site. Error bars indicating standard errors.**

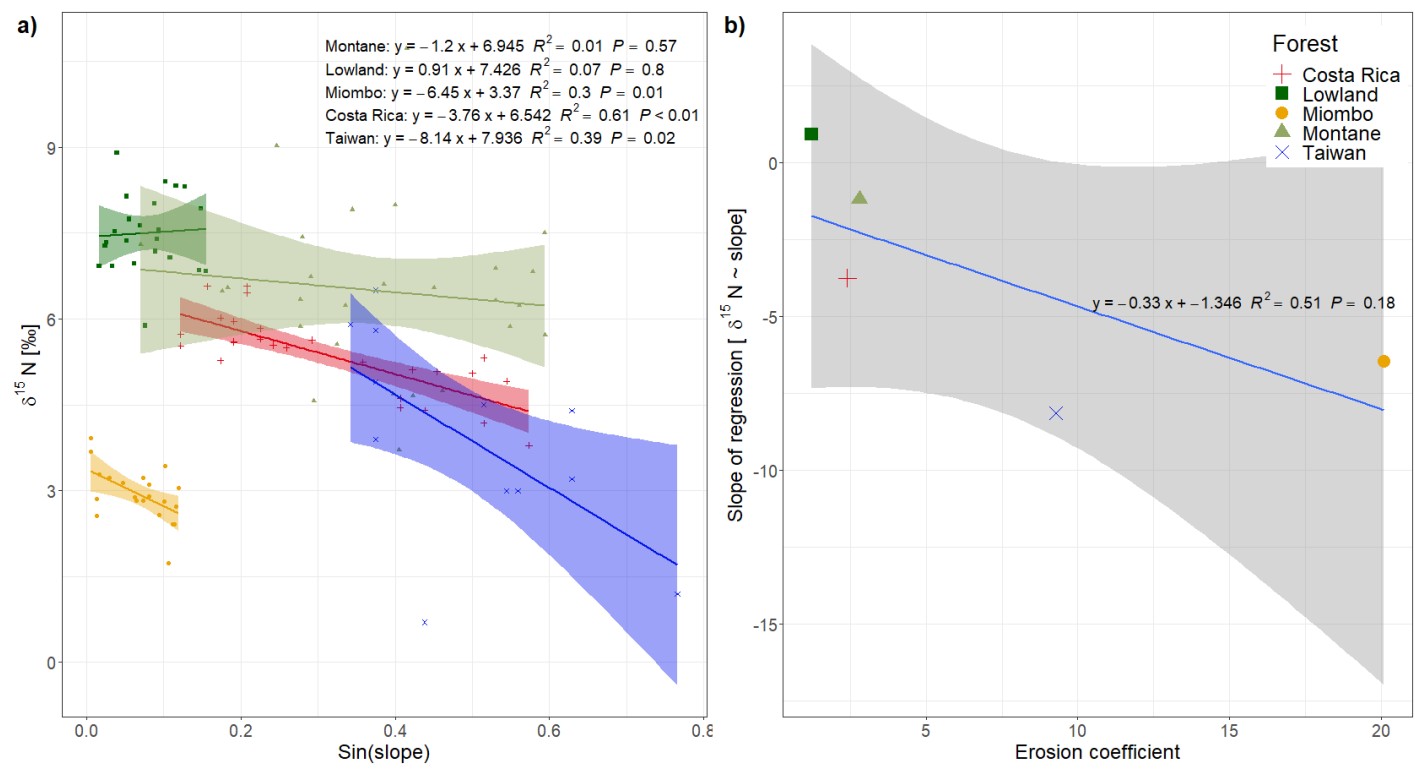

**Figure 3: a)** Soil δ¹⁵N values vs sinus of the slope from the spatial sampling of topsoil (0-20 cm) in DRC, including data from Costa Rica (Weintraub et al., 2015) and Taiwan (Hilton et al., 2013); **b)** slope of the regressions from panel (a) plotted against the mean erosion coefficient (Eq. 2) of each study site.

## 4 Discussion

### 4.1 Using soil δ¹⁵N to assess differences in ecosystem N-turnover

At present, relatively little information on δ¹⁵N data for soils in tropical forests is available. This holds especially for Central Africa, where almost no data is reported. Rütting et al. (2015) is one of the few studies that reported on δ¹⁵N from montane tropical forests in Rwanda. Their values for topsoil (0-5 cm) range between 3.6 and 4.6 ‰ and are similar to the values reported in this study for the montane forest (mean of 4.4 ‰ for 0-6 cm depth in this study) and are also in the same range reported from topsoil in Rwanda by Bauters et al. (2017). Lower values were found in an old-growth montane forest topsoil (0-10 cm) in Ethiopia (-3 to 1 ‰, Eshetu and Högberg, 2000) and in semi-natural montane forests in Tanzania (0-4 ‰, Gerschlauer et al., 2019). δ¹⁵N values from tropical mountain forests of Taiwan are also in the same range, with a mean value of 4 ‰ for the upper 10 cm of the soil profile (Hilton et al., 2013). Jamaican soils from tropical montane forests are much more depleted in ¹⁵N, with values from -1.6 to 1.5 ‰ for depths down to 20 cm (Brearley, 2013). Top soils from an altitudinal gradient in Borneo were also more depleted, with a mean value of 1.6 ‰ (Kitayama and Iwamoto, 2001). δ¹⁵N values reported for tropical lowland forests are available for soils in Costa Rica where top soils (0-10 cm) showed values of 4.7 ‰ (Osborne et al., 2017) and a

mean value of 5.9 ‰ for soils from 0-20 cm depth (Weintraub et al., 2015). Higher erosion, due to high rainfall and uplift rates (1.7-8.5 mm yr$^{-1}$), may explain why these values of lowland forest in Costa Rica are more depleted in the heavier N isotope compared to the lowland forest in Yoko from this study (7.55 $\pm$ 0.69 ‰ for 0 – 20 cm soil samples) because the residence time
of soil N decreases with erosion and hence soil rejuvenation (Amundson et al., 2003). The $\delta^{15}$N reported here for the Miombo woodland (2.68 – 4.32 ‰ between 0 – 20 cm, Figure 2) falls within the same range as values reported by Wang et al. (2013) from a moist woodland savanna in Zambia (2.6 – 4.8 ‰) and a mature Miombo woodland in Tanzania (3.5 – 3.8 ‰, Mayes et al., 2019), but are lower compared to values reported from a dry forest in the Cerrado Savanna of Brazil (6.3 – 10.8 ‰, Bustamante et al., 2004).

Soil $\delta^{15}$N is related to the residence time of ecosystem N. Higher rates of mineralization, nitrification and dentification leaves enriched N in the remaining soil organic matter (SOM), while the more depleted mineral N is removed via plant uptake or denitrification and transported down into deeper soil layers by leaching (Handley et al., 1999; Hobbie and Ouimette, 2009). Over the whole profile, the Miombo woodland showed highest depletion of $^{15}$N (Figure 2), which would indicate a slower N turnover and a more closed N cycle compared to the lowland and montane forest. However, as the $\delta^{15}$N signatures are also
influenced by soil N inputs, the Miombo woodland, dominated by the legume tree *Brachystegia*, likely receives relatively more depleted $^{15}$N through biological N$_2$-fixation. Previous studies have shown that N$_2$-fixation is a more important process in tropical woodlands than in tropical forests (Hogberg & Alexander 1995), where less trees of the Fabacaea family are present (Malmer and Nyberg, 2008). Furthermore, it is reported that symbiotic N$_2$-fixation is downregulated in old-growth tropical lowland forest (Bauters et al., 2016), thus $\delta^{15}$N of top soil in these forests is less likely to be affected by N$_2$-fixation, which is
clearly indicated by more enriched top soil $\delta^{15}$N values. Beside N$_2$-fixation, topsoil $\delta^{15}$N is also influenced by the isotopic signature of incoming litter fall, inputs through roots and N deposition (Craine et al., 2015b), while downward transport processes and transformation of SOM influences the $\delta^{15}$N in deeper soil layers (Baisden et al., 2002). In this study, the Miombo woodland showed slightly lower $\delta^{15}$N values for the topsoil (0-2 cm, 2.66 ‰) than the montane forest (2.94 ‰), while the lowland forest showed a more enriched signature (6.35 ‰). The foliar $\delta^{15}$N presented in Table 3 show that the values in the
Miombo and montane forest are substantially lower than in the lowland forest and that $\delta^{15}$N inputs from plants lower the soil $\delta^{15}$N values in these forests more compared to the lowland forest. Whereas the strong depletion of the Miombo woodland can be related to atmospheric N$_2$-fixation, the shift of plant and topsoil $\delta^{15}$N to more depleted values in the montane forest could also be an altitudinal effect. Depletion of $\delta^{15}$N with increasing elevation in tropical forests, have been observed in the netropical and afrotopical forests and were linked to a more conservative N cycle at high altitudes (Bauters et al., 2017).

Soil $\delta^{15}$N profiles of the lowland forest and the Miombo woodland showed a $\delta^{15}$N enrichment peak at 10 cm and 30 cm depths (Figure 2b), respectively, indicating the highest microbial activity and/or leaching of soil N in this layer. Montane forest showed a similar peak at the same depth, indicating high turnover or leaching in this soil layer as well. However, from 20 cm onwards $\delta^{15}$N enrichment increased even further to the highest values at 100 cm depth (Figure 2b). Hobbie & Ouimette (2009) presented a conceptual model for processes that influence $\delta^{15}$N values in a profile. According to this model, a profile with
higher $\delta^{15}$N values in the root layers and decreasing $\delta^{15}$N in subsequent depths (similar to the lowland and Miombo profiles)

is more likely to occur in an open system with lower N limitation and more inorganic N cycling (Hobbie and Ouimette, 2009). Denitrification and associated gaseous N losses cause the increase of soil $\delta^{15}N$ at intermediate depth. Leaching and subsequent microbial assimilation of depleted $NO_3$ at lower depths result in the more depleted $\delta^{15}N$ signature of lower soil layers compared to the intermediate depth (Hobbie and Ouimette, 2009). This is also in line with the higher $N_2O$ fluxes that were observed in the lowland forests of Yoko and in Miombo woodlands in Tanzania compared to montane forests in Kahuzi-Biéga (Table 3), indicating generally a higher N turnover (especially nitrification and denitrification) occurring at these sites. Moreover, Bauters et al. (2019) showed indication of mineral N excess in the Yoko lowland forest, which is also due to high atmospheric N depositions in the central African lowlands (Table 3). Overall, the soil $\delta^{15}N$ profiles of the lowland forest and the Miombo woodland seem to have a more open N cycle due to excess N availability, which is most likely due to high N depositions in the lowland and high $N_2$-fixation in the Miombo.

On the other hand, a soil profile where $^{15}N$ continuously increases with depth (i.e. the case of the montane forest) is indicative of an N-limited ecosystem or dominated by organic N cycling. In these systems, assimilation of leachate (depleted in $\delta^{15}N$) in deeper soil-layers is less common, thus the $\delta^{15}N$ steadily increases with depth. Although, we found highest N stocks in the montane forest, higher foliar C:N ratios of tropical forests with increasing altitude (Bauters et al., 2017) indicate that this ecosystem experiences lower N availability and is dominated by organic N cycling. In Rwandan and Ecuadorian forests, Bauters et al. (2017) concluded from leaf $\delta^{15}N$ altitudinal gradients that N-cycling becomes more closed with increasing elevation and hypothesized that organic N sources for plants become more important in ecosystems with lower temperatures. Furthermore, very high mineralization rates have been measured in the montane forests of Kahuzi-Biéga, which indicates a high organic matter turnover in these forests (Bauters et al., 2019) which is supported by the high aquatic TDN and DON exports from this forest (Table 3).

Table 2: Fixed effects estimates for $\delta^{15}N$, N-stocks and C:N ratios. Fixed effects include forest type (Lowland, Montane and Miombo) and soil depth in cm (for $\delta^{15}N$ and C:N). N-stocks are for top 100 cm of the soils. For each effect, estimated standard error and estimated P-values is given, along with the estimated marginal (m) and conditional (c) $R^2adj$ (Nakagawa and Schielzeth, 2013).

| Response | Effect | Effect size | SE | P-value | $R^2_{adj,m}$ | $R^2_{adj,c}$ |
|---|---|---|---|---|---|---|
| $\delta^{15}N$ [‰] | Intercept: Lowland | 7.72 | 0.10 | < 0.001 | 0.65 | 0.65 |
| | Montane | -1.82 | 0.18 | < 0.001 | | |
| | Miombo | -4.08 | 0.16 | < 0.001 | | |
| | depth | -0.002 | 0.00 | 0.26 | | |
| C:N | Intercept: Lowland | 7.96 | 0.16 | < 0.001 | 0.57 | 0.58 |
| | Montane | 3.94 | 0.26 | < 0.001 | | |
| | Miombo | 4.20 | 0.22 | < 0.001 | | |
| | depth | -0.02 | 0.00 | < 0.001 | | |
| N [kg m$^{-2}$] | Intercept: Lowland | 1.47 | 0.17 | < 0.001 | 0.36 | 0.36 |

| | | | |
|---|---|---|---|
| Montane | 0.65 | 0.27 | 0.160 |
| Miombo | -0.35 | 0.24 | 0.03 |


**Table 3: Foliar plant $\delta^{15}$N [‰] values, N deposition rates [kg N ha$^{-1}$ yr$^{-1}$], aquatic TDN and DON losses [kg N ha$^{-1}$ yr$^{-1}$] for the three study sites. Aquatic TDN losses and DON losses are preliminary results from measurements by the authors of this study at the respective study sites (methods see Supplement).**

| | Lowland | Montane | Miombo |
|---|---|---|---|
| Foliar $\delta^{15}$N [‰] | 3.42 | 0.58 | -0.74 [1] |
| N deposition [kg N ha$^{-1}$ yr$^{-1}$] | 18.2 [2] | 21.2 [2] | 3-5 [1] |
| Aquatic TDN loss [kg N ha$^{-1}$ yr$^{-1}$] | 3.87 | 6.50 | 0.84 |
| Aquatic DON loss [kg N ha$^{-1}$ yr$^{-1}$] | 2.37 | 4.46 | 0.73 |
| $N_2O$-flux [kg N ha$^{-1}$ yr$^{-1}$] | 1.38 [3] | 0.59 [3] | 1.16 [4] |

*References: 1) Mayes et al., 2019 (Miombo woodland Tanzania); 2) Bauters et al., 2019 (Lowland forest in Yoko, DRC;*
*Montane forest in Kahuzi-Biéga, DRC); 3) Barthel et al., 2021 (in review); 4) Rees et al., 2006 (Miombo Woodland Tanzania)*

## 4.2 Influence of topography on soil $\delta^{15}$N

A possible effect of topography on soil $\delta^{15}$N is most likely due to higher erosion of steeper slopes compared to flat landscapes, because we assume that within a site other factors controlling soil $\delta^{15}$N (such as temperature, precipitation and vegetation
cover) do not vary with topography. In contrast to the Costa Rican tropical lowland forests (Weintraub et al., 2015), slope angle was found to have no effect on $\delta^{15}$N in the lowland forest of this study (Figure 3a). In contrast to the Costa Rican site, there are no steep slopes in the lowland forest of this study (max. slope sampled 9°) and thus, there is a low potential for physical erosion. This is also indicated by a lower EC (1.22) in the lowlands compared to Costa Rica (2.39) (Table 1). Although the Miombo woodland is located in an area, with similar topography to the lowland forest (Figure 1), the EC from this area is
much higher (20.11). This can be explained by the fact that the forest cover in the Miombo is less dense than the tropical forests and this increases the exposure of the soil surface to the energetic input of rainfall. Furthermore, the loss of understory vegetation from fire, which occurs frequently during the dry season (Campbell, 1996), leaves the soil unprotected from erosion at the onset of the rainy season. The spatial topsoil samples of the Miombo woodland clearly indicate a significant effect of slope angles on $\delta^{15}$N (Figure 3a). The $\delta^{15}$N values in the montane forest, situated on steep terrain, tend to decrease with
increasing slope angle, however, the visible trend is not statistically significant (*P*-value of 0.57, Figure 3a). Furthermore, we found a higher variability in $\delta^{15}$N values of the soil samples in the montane forest (CV of 22 %) compared to the Miombo (CV of 15 %) and lowland forest (CV of 9 %). While slope angle is a commonly used predictor of soil erosion potential, the curvature of the land surface (i.e. convex or concave) has also been shown to affect soil erosion at a local scale (Stefano et al., 2000). The specific hydrology seems to control the erosive mechanism, as slope gradients have been found to influence the

rate of soil erosion through overland flow, while curvatures affect diffusive processes, such as splash erosion (Stefano et al., 2000). However, while a trend of lower $\delta^{15}$N on more convex landscapes in the montane forest is visible (Figure S1), no statistical significance was found either, as samples were not equally distributed over the range of curvatures. In addition, because our sampling strategy was focused on the slopes, sample sites were biased towards concave curvatures.

Comparing the effect of slope angles on topsoil $\delta^{15}$N of the study sites from the Congo Basin with the effect observed in
tropical forests of Costa Rica (Weintraub et al., 2015) and Taiwan (Hilton et al., 2013) shows that the magnitude of the effect is related to the mean erosion coefficient in these ecosystems (Figure 3b). Sites with a high erosion coefficient show a higher slope of the regression $\delta^{15}N \sim slope\ angles$ compared to sites with a low erosion coefficient, implying a bigger effect on soil $\delta^{15}$N in more geomorphic active sites. As the dense tropical lowland forests of the Congo Basin do not represent high EC values, the local effect of erosion on soil $\delta^{15}$N seems negligible. Although the montane forest shows a moderate EC, due to the
steep topography, the high variability in soil N likely masks the potential effect of topography on $\delta^{15}$N in our dataset. The estimation of EC based on equation 1 assumes that splash and overland flow are the dominant erosion processes but this may underestimate erosional forces at higher hillslope angles (Pelletier, 2012). At higher hillslope angles, landslides will be more important erosional processes than overland flow. This might indicate that at sites with high risks of landslides (as shown in the landslide hazard index (LSI) in Table 1) are even more prone to physical soil loss than the EC would indicate. In this case
especially the forest in Taiwan (LSI of 4), the montane forest (LSI of 3.96) and the forest in Costa Rica (LSI of 3.65) experience a high risk of landslides and total soil loss in this forest might even be higher than predicted by the EC.

We applied a SEM to assess the driving factors of the topsoil $\delta^{15}$N variability between different tropical forest sites (Figure 4). On a global scale, soil $\delta^{15}$N is reported to be mainly influenced by climatic factors (MAT and MAP) and soil properties, such as soil C and clay content (Craine et al., 2015a). Our results showed that soil C had only a marginal influence (lower coefficient
compared to slope, MAP and LAI) on tropical soil $\delta^{15}$N and is only significantly influenced by MAT and not MAP. Soil $\delta^{15}$N in these tropical forests seems to be primarily influenced by factors controlling erosion (MAP, LAI and slope). Beside the direct effect of MAP on soil $\delta^{15}$N, there is also a significant indirect effect visible, where sites with higher MAP experience also higher LAI values (Figure 4). It was not possible to analyze the effect of clay content on $\delta^{15}$N because the data were not available for all the sites. Although our data showed only for the Miombo a significant effect of topography on the soil $\delta^{15}$N
values, our pantropical analysis suggests that especially in geomorphic active regions, erosion needs to be taken into account to explain local differences in soil $\delta^{15}$N values. However, it is important to state that the SEM with only 112 samples from 5 different ecosystems represents an overly simple model and more data is needed for a more precise outcome. Unfortunately, additional $\delta^{15}$N data, that includes topographical variability is not available, to the best of our knowledge.

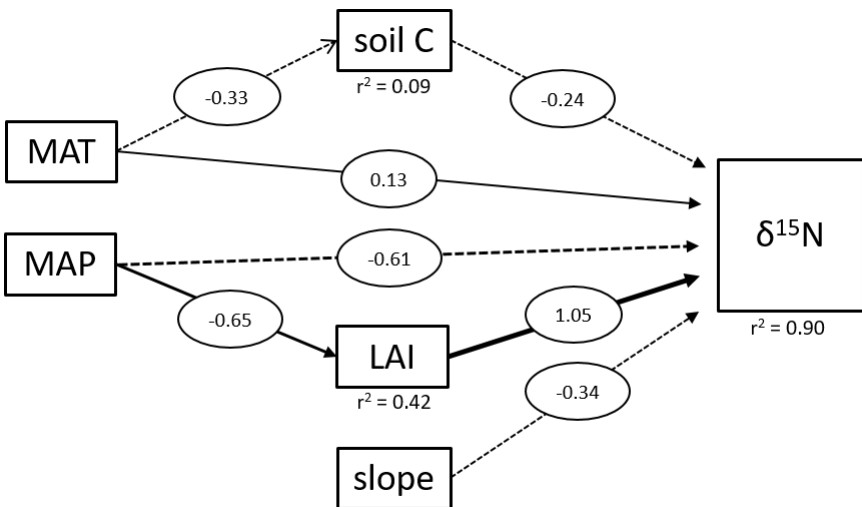

**Figure 4: Structural equation model of direct and indirect effects on tropical soil δ¹⁵N values. The diagram shows all significant relationships. MAT represents the mean annual temperature in °C, MAP the mean annual precipitation in mm and LAI the leaf area index. Soil C is the carbon content of the soil samples and slope the slope angles of the sampling site. Numbers represent the strength of each effect.**

## 5 Conclusion

The three tropical forest ecosystems in the Congo Basin each showed a distinct $\delta^{15}$N soil profile. The $\delta^{15}$N soil profiles in the montane forest indicate a closed N cycle, which supports the common theory of decreasing N availability at higher altitudes. In contrast, the $\delta^{15}$N soil profiles indicated that the lowland forest and Miombo woodland tended to have more open N cycles, which were mostly dominated by inorganic N cycling. The Miombo woodland showed the lowest $\delta^{15}$N values, which is most plausibly explained by enhanced $N_2$-fixation due to the tree species composition dominated by the legume *Brachystegia*.

Furthermore, topsoil $\delta^{15}$N signatures across our and literature sites of the tropics suggests that a significant amount of the observed variability can be explained by erosional processes. Rainfall, vegetation cover, and topography are the main factors to explain $\delta^{15}$N variability between five different tropical forest sites. Within the Congo Basin, only the Miombo woodland showed an influence of slope on soil $\delta^{15}$N, whereas in the lowland forest, the comparably low MAP, dense vegetation, and flat topography likely limit erosion and resulted in the lack of a correlation between the $\delta^{15}$N of soil N and slope angle. In general,

this study highlights the possible importance of soil erosion on N budget in tropical forests However, not all sites are similarly impacted. Therefore, soil erosion has to be taken into account while interpreting of soil $\delta^{15}$N values as a proxy for N cycling. It underlines the importance of further spatial field studies in the tropics given the observed high variabilities between different tropical forest types.

## Data availability

All data used in this study were published at Zenodo and are available under http://doi.org/10.5281/zenodo.4113895.

**Authors contribution:**

SB, MBauters, PB and KvO designed the research project. SB and MBauters carried out field- and laboratory work. Data analysis and interpretation was performed by SB, MBauters, KvO, TWD, MBarthel, JS and PB. SB wrote the manuscript with contributions from all co-authors.

**Competing interests:**

The authors declare no conflict of interest.

**Acknowledgments:**

We thank the authorities and park rangers of the *Institute Congolais pour la Conservation de la Nature* (ICCN) for the help in accessing the field site in the Kahuzi-Biéga National Park. We are grateful for Serge Alebadwa, Nadine Bahizire, Claudino
Sumaili, Degra Ngoy, Christ Ibrahimu and Rosine Bakengere for assistance during the fieldwork. We are also thankful for Katja Van Nieuland and Samuel Bodé for the help with sample analysis. This research has been funded by the Fonds de la Recherche Scientifique FNRS project numbers T.0059.18 and J.0167.19.

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
