# Peer review of "Figure S1: $\delta^{15}\text{N}$ in topsoil (0 – 20 cm) plotted against the curvature of the sampling spot. Positive curvature values represent concave landforms, while negative values represent convex landforms."

_SOIL, 2020_

## Referee Comment (RC1) · Anonymous Referee #1 · 17 Dec 2020

General comments The study used $\delta$15N of soil profiles to assess ecosystem-level differences in N cycling in three forest ecotypes within the Congo Basin (tropical lowland forest, tropical montane forest, and subtropical Miombo woodland). Based on the distinct $\delta$15N soil profile observed in each forest, the authors conclude that the montane forest indicate a closed N cycle the lowland forest and Miombo woodland tended to have more open N cycles. The study also examined the effect of surface slope angles on $\delta$15N in the same forests to quantify local differences induced by topography, but they found a contrasting effect. Furthermore, the study did a pan-tropical analysis of soil $\delta$15N to reveal that rainfall, vegetation cover, and topography are the main factors to explain $\delta$15N variability between five different tropical forest sites. I find the submission

to be well-written and relatively thorough with valuable contribution to the literature on N cycling in tropical forests, for which limited information is available. The subject of the study is suitable for SOIL. However, there are some conceptual and technical problems and manuscript should be revised before it is accepted. General comments Although the study briefly mentioned that soil $\delta$15N values can signal openness of ecosystem N cycle (line 68-69), it lacks explanation on how soil $\delta$15N values are interpreted as integrator of N cycling. Indeed, the interpretation of soil (and plant) $\delta$15N values as indicators of N availability is not straight forward with many contradicting interpretations of observed pattern of soil/plant $\delta$15N, and this need to be highlighted in the study with relevant studies from local to global scales. Many factors (not only N availability) affect soil $\delta$15N values at a given site and across sites. Particularly, I am concerned with the lack of data on plant áž§15N. There is no linearity between soil $\delta$15N values and N viability, and this needs to be acknowledged in the manuscript in depth, which is also supported by the data in this study. Another major issues/question is why only five sites are included in the SEM? As a result, the SEM was also overly simplified (few sites and few potential variables). Some relevant studies from the bulk studies in other tropical forests in Africa, SE Asia, and S America should be included in the analysis as well as discussion of the results in this study. Specific comments Line 16: Change 'stable isotope signature' to 'natural abundance of stable 15N isotope' Line 23: 'no influence of topography on soil N cycling'. This is not supported by the study. No effect of topography on soil $\delta$15N does not mean that topography has no effect on soil N cycle, which is broader than soil $\delta$15N. The author needs to be more cautious when using N cycling instead of soil $\delta$15N. Line 24: '$\delta$15N' needs to be referred to consistently (soil $\delta$15N, soil $\delta$15N signature, stable isotope signature…are all used to refer to soil $\delta$15N in the manuscript). Line 35: 'However' does not seem necessary Line 44: replace 'forest' by 'forests' Line 52: Delete 'activity' Line 55: Correct 'intact' as 'an intact' Line 58-62: revise these sentences. Consider this 'Some studies from geometrically active sites of the tropics (Costa Rica and Taiwan) found lower N availability and more closed N cycle in steeper sloping positions suggesting that erosion has a significant control on

N cycling (Hilton et al., 2013; Weintraub et al., 2015). However, and the magnitude of this effect in more stable landscapes is unknown calling for a consistent study across geomorphic gradients in the tropics. Line 64 : Edits 'The stable isotope composition of N (áž§15N)' as 'The natural abundance of stable 15N isotope (áž§15N) of plant and soil pools' Line 66: should be 'insights into' Line 75-82: A testable hypothesis about the pattern soil áž§15N and N availability and openness of N cycle is needed. I would also question the hypothesis that soil $\delta$15N would be lower on steeper slopes because the erosion on steeper slopes removes fresh organic matter input from plants, which would continuously keep $\delta$15N of surface soil low compared to the deeper surface. Line 93: Are both forests used in this study? Line 130: 'Laboratory' is more formal Line 135: provide áž§15N of the atmospheric N2 Line 146-147: Why only these two sites were chosen? Line 155 (last sentence): Consider putting it at the end of the paragraph Line 157-159: The SEM analysis was very simple with only five sites with only few potential factors that affect soil áž§15N being included in the model. What are the variables included in the model? Line 161-165: The values of these variables needs to be directly presented; it is not enthusiastic to many readers to extract the information from the Table (estimates). Line 187: I would not use 'N cycling'. This study did not investigate the many aspects of N cycling. More importantly, the many factors known to affect soil áž§15N and which are very important to interpret soil áž§15N are not measured. Line 188-89: Eshetu et al., 2004 Forest Ecology and Management 187, 139–147 (Ethiopia) and Gerschlueret et al., 2019 Biogeosciences 16, 409–424 (Tanzania) are some of the relevant references missing. Line 207-208: This is not necessarily true as lower soil/plant áž§15N is not always associated with limited N availability (closed N cycle). Gurmesa et al., 2017 Biogeosciences, 14, 2359–2370 (many other studies in SE Asia) have reported ecosystems pools can be strongly 15-depleted under N saturated condition. Line 209: how about the effect of áž§15N of deposition N? Craine et al., 2015b? Line 214: depleted N-input from where? Only biological N deposition? Do you have data for N2-fixing plant species as well as their mycorrhizal association in the three forests? These are very crucial to interpret soil áž§15N values. Line 236: this

sentence does not help with the logical flow points being discussed in the paragraph Lines 237-238: Line 226-227 repeated? Again, as I mentioned above, low soil áž§15N does not necessary indicate closed N cycle. The context needs to be discussed. To say whether N cycle is dominated by organic N, it needs additional measurement. Is there data for soil inorganic N concentration in each forest? Line 239: Edit 'excess of available N' as 'excess N availability'. However, it is not correct to conclude that the forests have excess N availability only based on the values of soil áž§15N. Line 240: It is amazing that the author did not provide data on N deposition for any of the sites (including those from literatures). Line 248: change 'soil N' to 'soil áž§15N'. the discussion about effects of topography on soil áž§15N is interesting, but it did not establish mechanistic relationship of topography with other factors known to strongly affect soil áž§15N. The implication in discussion here is that soil áž§15N is strongly affected by physical process (erosion) and the factors that control the erosion. Line 289: 'samples' or 'sites?

Few technical corrections /writing Line 19: delete one of the 'in's Line 65: Should be Craine et al., 2015a. Also check line 209. Figure 2: first letter in y-axis label should be capitalized Figure 3: first letter in x-axis label should capitalized Table 2: Is it important to have all those decimals for fixed effect Estimates? References Clarke et al., 2013 (Line 32) and Vitousek 1985 (line 40) are missing. The superscript in 15N or áž§15N are not correctly written for many reference

---

## Referee Comment (RC2) · Anonymous Referee #2 · 8 Jan 2021

The manuscript quantifies the nitrogen (N) stocks and N isotopic composition of soils at three locations in the Congo Basin. The aim was to explore N availability in ecosystems across this poorly studied region, in the broader context of understanding N cycling in tropical forests. As a key macronutrient, the N cycle of these forests is a critical part of understanding how an ecosystem might respond to external drivers (changes in pCO2, climate, landuse). The study finds large contrasts in the stable N isotopic composition (d15N) between the sites, alongside changes in N stock, and seeks to link these to differences in environmental and geomorphic variables. At each site, the work explores how slope angle (and topographic position) influence d15N, building on some past work in Taiwan and Costa Rica, to explore how geomorphic processes influence

[Figure]

N cycling. The study was well focused, succinct, and the theme makes it worthy of attention at SOIL.

However, I found the discussion quite hard to follow, and it was hard to draw out the main findings. My main comments below reflect this, and make some suggestions for revisions:

1) Provide a clearer assessment of the potential controls on d15N in soil: This doesn't have to be more than a paragraph, as this has been done in other papers (from time to time), but the paper lacks a clear explanation of what controls the d15N values of soil N. This would be useful in the introduction, and then used to seed the structure of the discussion and help a clearer assessment of what best explains the patterns in the data. I would suggest something that talks about N inputs (and their d15N values), internal N cycling (plant to soil) and role of N losses (gaseous, dissolved, particulate) and how they may fractionate (or not) N isotopes in soil. Some of this is there in the manuscript, but its not that clear, and confused by the "open" vs "closed" discussion (see next point).

2) The "open" vs "closed" explanation for d15N values: This seems too simplified now, as we recognise that we can vary several aspects of the N cycle in an ecosystem and arrive at the same d15N values. For instance: i) the comparison between the N stock (N/km2) and input and output fluxes (N/km2/yr) can play a role, as with any isotope mass balance; ii) the N inputs (deposition, fixation) can be fractionated (or not); iii) the N outputs (gaseous, dissolved, particulate) can be fractionated (or not); iv) and pedogenesis and timescales of soil formation can vary (giving different intergration periods for different sites, and over depth). So with this explanation at hand, the simple argument of closed vs open is simplistic. In fact, the open vs closed model (I think) implicitly assumes that all N losses are fractionating, and that the ratio of N stock to N fluxes are the same at every site. Both those assumptions are flawed.

Instead, this study measures N stocks (and C/N, so relative to C). So it can say some-
thing about how this varies (and the paper doesn't use this information paired to the d15N data).

The study doesn't measure plant d15N, NO3 in porewaters or streams, or any gaseous N (that is very rare to do). This means any discussion of these important features of the N cycle and their d15N values would have to be drawn from other studies, and somewhat speculative for these sites. However, at the moment the paper doesn't discuss at all what these could be, and whether they could vary between the sites. By way of example, the lower MAP at the Miombo site could influence soil moisture – which is important for gaseous N loss (under saturated conditions) and NO3 loss (which can have a low d15N value). Thus, this could explain the shift in isotopic values: this site has less fractionating N losses. Or could it be simply a plant input (fixation) story.

Another quick example, the montane and lowland sites have similar d15N values, but the lowland site has much lower N stock (but higher relative to carbon C/N). So, to get the same d15N depletion in the soil residue, one has to invoke that the N fluxes out of the system (which fractionate) are larger in the montane system, than the lowland (because to see a d15N shift, you need the flux to be larger).

This text from me is somewhat off the top of my head. I could be completely off the mark here. But my point is that there are details to the dataset which are not discussed clearly, and the open vs closed discussion constrains this discussion in my view. A more structured discussion (see below) could also help.

3) Discussion section: I would recommend restructuring this to either take a more site by site explanation of patterns. Or a process by process explanation of patterns (e.g. starting with potential N inputs – could these explain things; then differences in N stocks; then potential N outputs). This could help draw out the key take away messages a little better.

Note – only having completed my review did I then read the comments already posted in the discussion. I found myself in agreement with queries flagged by the other reviewer.

Other comments (with line number):

19: maybe avoid the word "profiles" here – as the reader could infer you're talking about a soil profile, with depth.

19-20: this sentence would be better linked to the variability in d15N values measured, and how they've been interpreted.

23: this sentence on montane forest was a little confusing following the preceeding sentences, and perhaps the order of information here needs to be revised.

44: can the sentence "it is important" be rephrased to better spell out what the knowledge gaps are?

46: the "openness" section of text. I wonder if you need a couple of sentences explaining the inputs of N to ecosystems, and the losses. And then the idea that the overall size of the pool and leakiness is conceptualised as open vs closed. This might be clearer to those not familiar with the N cycle in soils.

60: I partly agree with that statement... But there is an important detail - Hilton et al., don't invoke the open vs closed concept. Instead, they argue that the nature of the N loss varies with slope, and that physical erosion and export of organic N in solid form does not fractionate the N isotope pool. In that way, the isotope mass balance is different for sites on steeper slopes (N loss dominated by non-fractionating losses), vs shallower slopes which potentially have a greater role of fractionating N losses (dissolved N forms, N gas forms).

69: please expand on the "openness of the N cycle" comment.

70: yes this is exactly what I write above! I should have been patient. Anyhow, I think perhaps that means that the order and flow of content might need some edits here.

105: experimental design seems good – and impressive range of sites across this

SOILD
[Figure]

setting. A quick Q – do you know the bedrock geology and whether it varies (and whether it could contain N?).

Figure 1 – please add a note to the caption that the colours are elevation (I guess?) and perhaps make a note of the resolution of the DEMs shown here.

Table 1 – is there a typo here? The lowland forest has the highest mean slope (22degrees) – which doesn't seem to fit with what you have shown in the histograms of slopes in Figure 1.

135: briefly detail the external standards used to re-calibrate the d15N values and their precision etc.,

140: adapted or used?

138: a bit more context on why this model was selected would be useful.

Figure 3 B – how did you lump the sites together to get this erosion coefficient?

Section 4.1. – I found this hard to follow. There is some repetition of themes and information (especially in the final paragraph), and it was hard to take away the main discussion points the authors wanted to highlight. It might make sense to start with a discussion of the N inputs, and the top soil values (and their contrasts) and what that indicates about them. The discussion N outputs/internal cycling (and depth profiles) at each site. And try to draw together a somewhat coherent discussion. One of the striking things is how high the d15N values are in the lowland (and at depth in the montane) and I finished this section without a clear idea what that was being attributed to.

Table 2: I don't understand the "Estimate" values in this table, and struggle to follow what they refer too.

251 – "there are no steep slopes in the lowland forest" – this does suggest that Table 1 is incorrect.

255: more about the controls on the EC output would be useful – as to why Miombo is so much higher. And how you computed the EC values for the literature data. And how Figure 3B came about (and the assumptions and limitations associated with it).

301: this note on N fixation was not clearly discussed in the main text – see comment above on Section 4.1

---

## Author Comment (AC1) · 20 Jan 2021

**Response to the interactive comment by the anonymous referee # 1**

General comments. The study used δ15N of soil profiles to assess ecosystem-level differences in N cycling in three forest ecotypes within the Congo Basin (tropical lowland forest, tropical montane forest, and subtropical Miombo woodland). Based on the distinct δ 15N soil profile observed in each forest, the authors conclude that the montane forest indicate a closed N cycle the lowland forest and Miombo woodland tended to have more open N cycles. The study also examined the effect of surface slope angles on δ15N in the same forests to quantify local differences induced by topography, but they found a contrasting effect. Furthermore, the study did a pan-tropical analysis of soil δ15N to reveal that rainfall, vegetation cover, and topography are the main factors to explain δ15N variability between five different tropical forest sites. I find the submission to be well-written and relatively thorough with valuable contribution to the literature on N cycling in tropical forests, for which limited information is available. The subject of the study is suitable for SOIL. However, there are some conceptual and technical problems and manuscript should be revised before it is accepted.

We thank the reviewer for the constructive and thorough review. We addressed the points raised by the reviewer on a point-by-point basis below. We are happy to address the mentioned concerns to improve our manuscript and believe that this will greatly benefit to the new MS quality.

General comments
Although the study briefly mentioned that soil δ15N values can signal openness of ecosystem N cycle (line 68-69), it lacks explanation on how soil δ15N values are interpreted as integrator of N cycling. Indeed, the interpretation of soil (and plant) δ15N values as indicators of N availability is not straight forward with many contradicting interpretations of observed pattern of soil/plant δ15N, and this need to be highlighted in the study with relevant studies from local to global scales. Many factors (not only N availability) affect soil δ15N values at a given site and across sites. Particularly, I am concerned with the lack of data on plant δ15N. There is no linearity between soil δ15N values and N viability, and this needs to be acknowledged in the manuscript in depth, which is also supported by the data in this study.

We acknowledge the concern of the reviewer that interpreting only soil $\delta^{15}N$ values and try to conclude based on these values on the nutrient status of the soils is ambiguous. Unfortunately, measurements on different processes and plant $\delta^{15}N$ are not available for all the sites. Thus, our interpretations will remain somewhat speculative. Nevertheless, we agree with the reviewer that this issue needs to be addressed more in depth in our manuscript and we will add more context and better identify limitations and uncertainties.

Another major issues/question is why only five sites are included in the SEM? As a result, the SEM was also overly simplified (few sites and few potential variables). Some relevant studies from the bulk studies in other tropical forests in Africa, SE Asia, and S America should be included in the analysis as well as discussion of the results in this study.

We agree with the reviewer that only a few data points were included in the SEM. Our goal was, in addition to studies in the literature, to include factors controlling for erosion (slope, LAI and MAP) in the model and to see if these factors can explain local soil $\delta^{15}N$ variability. Global datasets (for example from Craine et al. 2015) do not contain this information, as slope and LAI of the sampling points are missing. Furthermore, the given GPS coordinates are not precise enough to extract slope values from DEMs for the literature values and within site variability would be neglected. Thus, we included only the studies in the SEM model, where all information was available, and we ended up with 112 samples from 5 different tropical forest ecosystems. We will scan the literature again to see if no more data is available and add the concern of the over simplified SEM into the discussion of the results.

Specific comments
Line 16: Change 'stable isotope signature' to 'natural abundance of stable 15N isotope'

This will be changed accordingly.

Line 23: 'no influence of topography on soil N cycling'. This is not supported by the study. No effect of topography on soil δ15N does not mean that topography has no effect on soil N cycle, which is broader than soil δ15N. The author needs to be more cautious when using N cycling instead of soil δ15N.

We thank the reviewer for pointing this out and we agree that the sentence needs rephrasing. We will make sure that the differences between soil δ15N and soil N cycle are clearer in the manuscript.

Line 24: 'δ15N' needs to be referred to consistently (soil δ15N, soil δ15N signature, stable isotope signature…are all used to refer to soil δ15N in the manuscript).

We will revise the manuscript accordingly to be more consistent in the naming of the δ15N values.

Line 35: 'However' does not seem necessary

We agree with the reviewer and will remove "However".

Line 44: replace 'forest' by 'forests'
Line 52: Delete 'activity'
Line 55: Correct 'intact' as 'an intact'

We thank the reviewer for the grammatical corrections and will amend the manuscript accordingly.

Line 58-62: revise these sentences. Consider this 'Some studies from geometrically active sites of the tropics (Costa Rica and Taiwan) found lower N availability and more closed N cycle in steeper sloping positions suggesting that erosion has a significant control on N cycling (Hilton et al., 2013; Weintraub et al., 2015). However, and the magnitude of this effect in more stable landscapes is unknown calling for a consistent study across geomorphic gradients in the tropics.

This sentence will be revised in the new version of the manuscript.

Line 64 : Edits 'The stable isotope composition of N (δ15N)' as 'The natural abundance of stable 15N isotope (δ15N) of plant and soil pools'
Line 66: should be 'insights into'

We thank the reviewer for pointing this out and will address this in the new MS version.

Line 75-82: A testable hypothesis about the pattern soil δ15N and N availability and openness of N cycle is needed. I would also question the hypothesis that soil δ15N would be lower on steeper slopes because the erosion on steeper slopes removes fresh organic matter input from plants, which would continuously keep δ15N of surface soil low compared to the deeper surface.

We will rephrase the hypothesis to: "We hypothesized N availability and openness of N cycle would be highest in lowland tropical forest, which is indicated by lower δ15N signatures." We hypothesized that the isotopic signature of topsoil N is more depleted in steeper slopes compared to the isotopic

signature of topsoil N in less steeper slopes and not compared to the deeper surface of the same profile. We will rephrase this hypothesis to avoid confusion.

Line 93: Are both forests used in this study?

The sampled lowland forest catchment (260 ha) consists of these two forest subtypes. As we had a randomly spatial sample coverage, it is most likely that soils from both sub-types have been sampled. However, we did not identify all tree species at each sampling location to determine if it is a monodominant or mixed forest. We will amend the text that it is clearer to the readers.

Line 130: 'Laboratory' is more formal

We agree with the reviewer and will change the sub header to "Laboratory analysis".

Line 135: provide δ15N of the atmospheric N2

This information will be added to the revised version of the manuscript.

Line 146-147: Why only these two sites were chosen?

As we focused on the effect of topography and soil erosion on the soil $\delta^{15}N$ signature, only literature data with reported soil slope values of the samples were considered. To the best of our knowledge no other studies had a sampling strategy with within site variation of slope angles.

Line 155 (last sentence): Consider putting it at the end of the paragraph

This will be changed accordingly.

Line 157-159: The SEM analysis was very simple with only five sites with only few potential factors that affect soil δ15N being included in the model. What are the variables included in the model?

As described above we only included soil $\delta^{15}N$ data, where slope angles of the samples were available, thus only 5 sites were included in the SEM. We included MAP, MAT, LAI, slope and soil C content as predictive variables for soil $\delta^{15}N$ in the model. We will include this information into the new version of the manuscript.

Line 161-165: The values of these variables needs to be directly presented; it is not enthusiastic to many readers to extract the information from the Table (estimates).

The values for N stocks and C:N ratios are already mentioned in the text for each site. We will also add the values for the $\delta^{15}N$ to the text, that is easier for the reader to extract this information.

Line 187: I would not use 'N cycling'. This study did not investigate the many aspects of N cycling. More importantly, the many factors known to affect soil δ15N and which are very important to interpret soil δ15N are not measured.

We agree that the title might be misleading and suggest to change it to: "Using soil $\delta^{15}N$ signatures to assess differences in ecosystem N-turnover"

Line 188-89: Eshetu et al., 2004 Forest Ecology and Management 187, 139–147 (Ethiopia) and Gerschlueret et al., 2019 Biogeosciences 16, 409–424 (Tanzania) are some of the relevant references missing.

We thank the reviewer for providing additional references for our manuscript. So far, we listed only references from old growth natural tropical forests. The suggested papers are from young-growth forests in Ethiopia and semi-natural montane forests in Tanzania. However, it still might be interesting to expand our literature values and we are considering including these references into our manuscript.

Line 207-208: This is not necessarily true as lower soil/plant δ15N is not always associated with limited N availability (closed N cycle). Gurmesa et al., 2017 Biogeosciences, 14, 2359–2370 (many other studies in SE Asia) have reported ecosystems pools can be strongly 15-depleted under N saturated condition.

We thank the reviewer for pointing this out and agree that it is not clear that the presented soil $\delta^{15}N$ profiles indicate a more closed N cycle compared to the other two forest systems. We pointed this out in the subsequent sentence of our manuscript that the generally lower $\delta^{15}N$ values are probably influenced by the isotopic signatures of the inputs. However, we think that we can rephrase this sentence better to acknowledge the concerns more, using the proposed literature.

Line 209: how about the effect of δ15N of deposition N? Craine et al., 2015b?

We agree with the reviewer that N deposition influences the isotopic signature of soil N and will include this with the provided reference in the revised manuscript.

Line 214: depleted N-input from where? Only biological N deposition? Do you have data for N2-fixing plant species as well as their mycorrhizal association in the three forests? These are very crucial to interpret soil δ15N values.

While for the Miombo forest the depleted N-input probably is mainly from more N2-fixing, the montane forest is more likely to receive depleted biological N input via deposition (Bauters et al., 2017). Unfortunately, we do not have data on N2-fixing species available for our sites, but it is well documented this process is more important in the subtropical woodlands, compared to the tropical forests (Hogberg & Alexander 1995)

Line 236: this sentence does not help with the logical flow points being discussed in the paragraph

We thank the reviewer for pointing this out and agree that this sentence is indeed out of place. We will remove this sentence in the revised manuscript.

Lines 237-238: Line 226-227 repeated? Again, as I mentioned above, low soil δ15N does not necessary indicate closed N cycle. The context needs to be discussed. To say whether N cycle is dominated by organic N, it needs additional measurement. Is there data for soil inorganic N concentration in each forest?

We agree that the whole paragraph contains too many repetitions. We will restructure the whole paragraph to have a better flow for the readers. We measured aquatic N exports for all the catchments and the montane forests exports slightly more dissolved organic N (67% of TDN is DON) than the lowland forest (61%). We will add this data to the manuscript.

Line 239: Edit 'excess of available N' as 'excess N availability'. However, it is not correct to conclude that the forests have excess N availability only based on the values of soil δ15N.

This will be changed accordingly in the manuscript.

Line 240: It is amazing that the author did not provide data on N deposition for any of the sites

(including those from literatures).

N deposition data from montane and lowland forest are available from the literature and will be presented in the new version of the manuscript. Unfortunately, to our knowledge, no N deposition values are available for the Miombo forest.

Line 248: change 'soil N' to 'soil δ15N'.

We will change this in the revised manuscript.

the discussion about effects of topography on soil δ15N is interesting, but it did not establish mechanistic relationship of topography with other factors known to strongly affect soil δ15N. The implication in discussion here is that soil δ15N is strongly affected by physical process (erosion) and the factors that control the erosion.

It is true that other factors than erosion influence soil $\delta^{15}$N (temperature, precipitation and vegetation cover). However, these factors did not vary within our sites and only the physical processes were potentially influenced by slope gradients. We suggest that we address this issue shortly at the beginning of the paragraph.

Line 289: 'samples' or 'sites?

We thank the reviewer for the attention to the detail. Sites is correct and this will be changed.

Few technical corrections /writing
Line 19: delete one of the 'in's
Line 65: Should be Craine et al., 2015a. Also check line 209. Figure 2: first letter in y-axis label should be capitalized Figure 3: first letter in x-axis label should capitalized Table 2: Is it important to have all those decimals for fixed effect Estimates? References Clarke et al., 2013 (Line 32) and Vitousek 1985 (line 40) are missing. The superscript in 15N or δ15N are not correctly written for many reference

We thank the reviewer for mentioning these technical and writing errors. We will amend all proposed changes in the new version of the manuscript.

---

## Author Comment (AC2) · 20 Jan 2021

**Response to the interactive comment by the anonymous referee # 2**

The manuscript quantifies the nitrogen (N) stocks and N isotopic composition of soils at three locations in the Congo Basin. The aim was to explore N availability in ecosystems across this poorly studied region, in the broader context of understanding N cycling in tropical forests. As a key macronutrient, the N cycle of these forests is a critical part of understanding how an ecosystem might respond to external drivers (changes in pCO2, climate, landuse). The study finds large contrasts in the stable N isotopic composition (d15N) between the sites, alongside changes in N stock, and seeks to link these to differences in environmental and geomorphic variables. At each site, the work explores how slope angle (and topographic position) influence d15N, building on some past work in Taiwan and Costa Rica, to explore how geomorphic processes influence N cycling. The study was well focused, succinct, and the theme makes it worthy of attention at SOIL. However, I found the discussion quite hard to follow, and it was hard to draw out the main findings. My main comments below reflect this, and make some suggestions for revisions:

We thank the reviewer for the insightful comments and constructive review. We address all the reviewer's comments below and believe that - through a revision of the manuscript- the MS quality will greatly improve.

1) Provide a clearer assessment of the potential controls on d15N in soil: This doesn't have to be more than a paragraph, as this has been done in other papers (from time to time), but the paper lacks a clear explanation of what controls the d15N values of soil N. This would be useful in the introduction, and then used to seed the structure of the discussion and help a clearer assessment of what best explains the patterns in the data. I would suggest something that talks about N inputs (and their d15N values), internal N cycling (plant to soil) and role of N losses (gaseous, dissolved, particulate) and how they may fractionate (or not) N isotopes in soil. Some of this is there in the manuscript, but its not that clear, and confused by the "open" vs "closed" discussion (see next point).

We thank the reviewer for raising this point. We agree that the description of the factors controlling soil $\delta^{15}N$ signatures would be well placed in the intro. We will lay this out in the intro and link back to that in the discussion. This will also help structuring the manuscript better, in retrospect.

2) The "open" vs "closed" explanation for d15N values: This seems too simplified now, as we recognise that we can vary several aspects of the N cycle in an ecosystem and arrive at the same d15N values. For instance: i) the comparison between the N stock (N/km2) and input and output fluxes (N/km2/yr) can play a role, as with any isotope mass balance; ii) the N inputs (deposition, fixation) can be fractionated (or not); iii) the N outputs (gaseous, dissolved, particulate) can be fractionated (or not); iv) and pedogenesis and timescales of soil formation can vary (giving different intergration periods for different sites, and over depth). So with this explanation at hand, the simple argument of closed vs open is simplistic. In fact, the open vs closed model (I think) implicitly assumes that all N losses are fractionating, and that the ratio of N stock to N fluxes are the same at every site. Both those assumptions are flawed. Instead, this study measures N stocks (and C/N, so relative to C). So it can say something about how this varies (and the paper doesn't use this information paired to the d15N data).

The 'open' vs 'closed' system approach is widely used in literature. It is one way to interpret the scarce data available. However, we agree that the explanation is far from perfect and all the points mentioned by the reviewer will also have an influence on the soil $\delta^{15}N$ values and need to be addressed accordingly in the discussion. We see that we need to improve the way of discussing our data and put more emphasis on all the possible processes influencing the measured soil $\delta^{15}N$ values.

The study doesn't measure plant d15N, NO3 in porewaters or streams, or any gaseous N (that is very rare to do). This means any discussion of these important features of the N cycle and their d15N

values would have to be drawn from other studies, and somewhat speculative for these sites. However, at the moment the paper doesn't discuss at all what these could be, and whether they could vary between the sites. By way of example, the lower MAP at the Miombo site could influence soil moisture – which is important for gaseous N loss (under saturated conditions) and NO3 loss (which can have a low d15N value). Thus, this could explain the shift in isotopic values: this site has less fractionating N losses. Or could it be simply a plant input (fixation) story. Another quick example, the montane and lowland sites have similar d15N values, but the lowland site has much lower N stock (but higher relative to carbon C/N). So, to get the same d15N depletion in the soil residue, one has to invoke that the N fluxes out of the system (which fractionate) are larger in the montane system, than the lowland (because to see a d15N shift, you need the flux to be larger). This text from me is somewhat off the top of my head. I could be completely off the mark here. But my point is that there are details to the dataset which are not discussed clearly, and the open vs closed discussion constrains this discussion in my view. A more structured discussion (see below) could also help.

We thank the reviewer for pointing out this unclarity, we highly appreciate this.
Indeed, we agree with the reviewer that a more detailed discussion about the different possible mechanisms which can influence $\delta^{15}$N signatures is needed. The more depleted values in the Miombo woodland is most likely due to more N2-fixation, occurring in this ecosystem, compared to the tropical forests. The stable isotopic signature of soil N in the montane forest is especially in topsoil a lot lower compared to the signature in the lowland forest, this is indicating that N inputs are depleted and/or less fractionating processes (or more erosion) are happening in the topsoil. However, the different shape of the $\delta^{15}$N profiles tend to indicate different N availabilities as the shape present in the montane forest is typical for an N-limited ecosystem (increasing values with depth) while the shape in the lowland forest is more typical in N-rich ecosystems (highest $\delta^{15}$N value in an intermediate depth).

3) Discussion section: I would recommend restructuring this to either take a more site by site explanation of patterns. Or a process by process explanation of patterns (e.g. starting with potential N inputs – could these explain things; then differences in N stocks; then potential N outputs). This could help draw out the key take away messages a little better.

To improve the structure, we are glad to apply the suggestion by the reviewer and structure the section 4.1 by discussing process by process and then explain how every process influences $\delta^{15}$N at each site.

Note – only having completed my review did I then read the comments already posted in the discussion. I found myself in agreement with queries flagged by the other reviewer.

We are glad that both reviewers are agreeing on the points mentioned. We addressed everything from reviewer 1 in a separate response.

Other comments (with line number):
19: maybe avoid the word "profiles" here – as the reader could infer you're talking about a soil profile, with depth.

We indeed mean the soil $\delta^{15}$N distribution over the depth of the soil profile.

19-20: this sentence would be better linked to the variability in d15N values measured, and how they've been interpreted.

We agree with the reviewer that this sentence needs to be moved up and we will do so in the revised manuscript.

23: this sentence on montane forest was a little confusing following the preceeding sentences, and perhaps the order of information here needs to be revised.

To make it more coherent we will change this sentence to: "Despite the steep topography, slope angles do not constrain soil $\delta^{15}N$ in the montane forests, although this ecosystem experiences high variability in the stable isotope signature."

44: can the sentence "it is important" be rephrased to better spell out what the knowledge gaps are?

With this sentence we wanted to emphasize that different tropical forests are highly variable in nutrient availabilities and that a generalization of tropical forests being rich in N and poor in P is not suitable. This statement summarizes the points made before in the paragraph and helps to reason our research.

46: the "openness" section of text. I wonder if you need a couple of sentences explaining the inputs of N to ecosystems, and the losses. And then the idea that the overall size of the pool and leakiness is conceptualised as open vs closed. This might be clearer to those not familiar with the N cycle in soils.

We agree that in this paragraph some more information on the different processes altering soil $\delta^{15}N$ signatures is needed. We then will use the open vs. closed conception to summarize all the input vs. output processes of the system.

60: I partly agree with that statement... But there is an important detail - Hilton et al., don't invoke the open vs closed concept. Instead, they argue that the nature of the N loss varies with slope, and that physical erosion and export of organic N in solid form does not fractionate the N isotope pool. In that way, the isotope mass balance is different for sites on steeper slopes (N loss dominated by non-fractionating losses), vs shallower slopes which potentially have a greater role of fractionating N losses (dissolved N forms, N gas forms).

We agree that the concept of open vs. closed systems might not fit best for the studies cited here. We suggest that we change this sentence. The effect of physical soil loss on the $\delta^{15}N$ signature is described later in the introduction and is acknowledged by the reviewer below.

69: please expand on the "openness of the N cycle" comment.

As already stated above, this whole paragraph will be re-written, and the open vs closed system will be described in more detail.

70: yes this is exactly what I write above! I should have been patient. Anyhow, I think perhaps that means that the order and flow of content might need some edits here.

We will edit the manuscript in line 60 and don't link the two cited studies with the open vs. closed concept.

105: experimental design seems good – and impressive range of sites across this setting. A quick Q – do you know the bedrock geology and whether it varies (and whether it could contain N?).

We thank the reviewer for acknowledging our experimental design and we will try to find information on the bedrock geology of our study sites.

Figure 1 – please add a note to the caption that the colours are elevation (I guess?) and perhaps make a note of the resolution of the DEMs shown here.

Indeed, the colors show different evaluations of the sampling sites. We will add this information and the DEM source to the figure caption.

Table 1 – is there a typo here? The lowland forest has the highest mean slope (22degrees) – which doesn't seem to fit with what you have shown in the histograms of slopes in Figure 1.

The observation of the reviewer is right, the values for the mean slopes of the lowland and montane sites are swapped. This will be changed in the new version of the manuscript

135: briefly detail the external standards used to re-calibrate the d15N values and their precision etc.,

We will add a more detailed description of the stable isotope measurement to the manuscript.

140: adapted or used?

We used the model calculation of Pelletier 2012 and we will change the text accordingly.

138: a bit more context on why this model was selected would be useful.

We assume that the reviewer wants more context on why the SEM model was selected. We used a structural equation model as possible dependencies between variables can be included. We will add more information to the manuscript.

Figure 3 B – how did you lump the sites together to get this erosion coefficient?

As described in equation 1, the EC is calculated from the slope, MAP and the LAI. To calculate average EC values for each site, the average slope of every single sample from each site was used to calculate the respective EC.

Section 4.1. – I found this hard to follow. There is some repetition of themes and information (especially in the final paragraph), and it was hard to take away the main discussion points the authors wanted to highlight. It might make sense to start with a discussion of the N inputs, and the top soil values (and their contrasts) and what that indicates about them. The discussion N outputs/internal cycling (and depth profiles) at each site. And try to draw together a somewhat coherent discussion. One of the striking things is how high the d15N values are in the lowland (and at depth in the montane) and I finished this section without a clear idea what that was being attributed to.

As already suggested above, we will restructure the discussion in section 4.1 and discuss process by process and their potential influence on the $\delta^{15}N$ signature at each site. The higher $\delta^{15}N$ values in the lowland topsoil is most likely to less depleted N inputs compared to the Miombo woodland (less N2-fixation). The montane topsoil in general experiences a lot more erosion compared to the lowland forest, thus the stable isotopic signature is more depleted compared to the lowland forest. The isotopic values of the $\delta^{15}N$ profile of the montane forest are steadily increasing with depth and this is a typical shape of profile of an N limited ecosystem (Hobbie and Ouimette, 2009). While rephrasing the whole paragraph, we will make sure that the underlying processes are more clear to the reader.

Table 2: I don't understand the "Estimate" values in this table, and struggle to follow what they refer too.

The values in table 2 show the results for all the fixed effects estimated by the linear-mixed effects model (Estimate) and the respective standard errors and P-values. A presentation like this lets the

reader easily find the mean values for each response variable and it is clearly visible which effects differ significantly from others. We will change the table from "Estimate" to "Effect size" to make it clearer for the readers

251 – "there are no steep slopes in the lowland forest" – this does suggest that Table 1 is incorrect.

As mentioned above the slope values in Table 1 are switched between montane and lowland forest and will be changed in the revised manuscript.

255: more about the controls on the EC output would be useful – as to why Miombo is so much higher. And how you computed the EC values for the literature data. And how Figure 3B came about (and the assumptions and limitations associated with it).

The driving variable behind the high EC value in the Miombo forest is mainly the low LAI, as we already stated in the text. The forest cover in the Miombo is less dense compare to the other forest ecosystems and thus the soil is less protected from the erosive force of rainfall events. In section 2.4 we described how we calculated the EC values and where we obtained the data for the values of the literature. However, we agree that we can describe the assumptions and limitations of the model more clearly in the text and will do so in the revised manuscript.

301: this note on N fixation was not clearly discussed in the main text – see comment above on Section 4.1

We discussed the possible influence of fixed $N_2$ on the $\delta^{15}N$ signature in the Miombo woodland in section 4.1 L 213-219. However, the whole section 4.1 will be re-structured as suggested and we will try to discuss this issue of N fixation more thoroughly.

---

## Editor Comment (EC1) · Peter Finke (Editor) · 22 Jan 2021

I thank both the reviewers and the authors for thorough comments and responses, and invite the authors to produce a revised version of their manuscript based on what they propose as modifications. Since there may be some "structural" changes in the discussion sections, a "track-changes" version of the revision would be useful to identify actions taken.

Regarding the Response from the authors I have the following remark: Ad "Response to the interactive comment by the anonymous referee # 2": In the response to the second part of Comment 2) the authors do not indicate if and how they would adapt

the manuscript based on these comments.

Peter Finke, handling topical editor.
* * *

---

## Author Response (AR1)

**Topical Editor Decision: Revision (22 Jan 2021) by Peter Finke**
Comments to the Author:
I thank both the reviewers and the authors for thorough comments and responses, and invite the authors to produce a revised version of their manuscript based on what they propose as modifications. Since there may be some "structural" changes in the discussion sections, a "track-changes" version of the revision would be useful to identify actions taken.

Regarding the Response from the authors I have the following remark:
Ad "Response to the interactive comment by the anonymous referee # 2":
In the response to the second part of Comment 2) the authors do not indicate if and how they would adapt the manuscript based on these comments.

We thank the editor and reviewers for their constructive feedback and present here our final responses to each point. We uploaded a track-changed version of the revised manuscript and provide line-numbers in this response letter. In this final response we also go into more detail on the second part of Comment 2) of the referee #2 and state how we changed the manuscript accordingly (see below).

**Response to the interactive comment by the anonymous referee # 1**

General comments. The study used δ15N of soil profiles to assess ecosystem-level differences in N cycling in three forest ecotypes within the Congo Basin (tropical lowland forest, tropical montane forest, and subtropical Miombo woodland). Based on the distinct δ 15N soil profile observed in each forest, the authors conclude that the montane forest indicate a closed N cycle the lowland forest and Miombo woodland tended to have more open N cycles. The study also examined the effect of surface slope angles on δ15N in the same forests to quantify local differences induced by topography, but they found a contrasting effect. Furthermore, the study did a pan-tropical analysis of soil δ15N to reveal that rainfall, vegetation cover, and topography are the main factors to explain δ15N variability between five different tropical forest sites. I find the submission to be well-written and relatively thorough with valuable contribution to the literature on N cycling in tropical forests, for which limited information is available. The subject of the study is suitable for SOIL. However, there are some conceptual and technical problems and manuscript should be revised before it is accepted.

We thank the reviewer for the constructive and thorough review. We addressed the points raised by the reviewer on a point-by-point basis below. We were happy to address the mentioned concerns to improve our manuscript and believe that this has greatly benefited to the new MS quality.

General comments
Although the study briefly mentioned that soil δ15N values can signal openness of ecosystem N cycle (line 68-69), it lacks explanation on how soil δ15N values are interpreted as integrator of N cycling. Indeed, the interpretation of soil (and plant) δ15N values as indicators of N availability is not straight forward with many contradicting interpretations of observed pattern of soil/plant δ15N, and this need to be highlighted in the study with relevant studies from local to global scales. Many factors (not only N availability) affect soil δ15N values at a given site and across sites. Particularly, I am concerned with the lack of data on plant δ15N. There is no linearity between soil δ15N values and N viability, and this needs to be acknowledged in the manuscript in depth, which is also supported by the data in this study.

We acknowledge the concern of the reviewer that interpreting only soil $\delta^{15}N$ values is not straight forward. We therefore included, as suggested, additional data of foliar $\delta^{15}N$ which we measured at the lowland and montane forest site and foliar $\delta^{15}N$ from a Miombo woodland in Tanzania from the literature (see new Table 3). We also revised the paragraph in the introduction about the controls on

soil $\delta^{15}$N values (lines 70– 82) to give a reader a more detailed insight on how $\delta^{15}$N values can be used as an integrator of N cycling. Furthermore, we restructured and extended the discussion in paragraph 4.1 to elaborate on the possible controls on soil $\delta^{15}$N. Together, we believe that these revisions provide a more nuanced view on how soil $\delta^{15}$N data can give clues for N availability.

Another major issues/question is why only five sites are included in the SEM? As a result, the SEM was also overly simplified (few sites and few potential variables). Some relevant studies from the bulk studies in other tropical forests in Africa, SE Asia, and S America should be included in the analysis as well as discussion of the results in this study.

We agree with the reviewer that only a few data points were included in the SEM. Our goal was, in addition to studies in the literature, to include factors controlling for erosion (slope, LAI and MAP) in the model and to see if these factors can explain local soil $\delta^{15}$N variability. Global datasets (for example from Craine et al. 2015) do not contain this information, as slope and LAI of the sampling points are missing. Furthermore, the reported coordinates are not sufficiently precise to extract slope values from DEMs for the literature values and within site variability would be neglected. Thus, we included only those studies in the SEM model, where all information was available; this resulted in 112 observations from 5 different tropical forest ecosystems. We added this concern to the method of the inter-site comparison (lines 176-179) and added two sentences at the end of the discussion paragraph stating the limitations of the SEM, with only five sites included (lines 363-365).

Specific comments
Line 16: Change 'stable isotope signature' to 'natural abundance of stable 15N isotope'

This has been changed accordingly (line 16).

Line 23: 'no influence of topography on soil N cycling'. This is not supported by the study. No effect of topography on soil δ15N does not mean that topography has no effect on soil N cycle, which is broader than soil δ15N. The author needs to be more cautious when using N cycling instead of soil δ15N.

We thank the reviewer for pointing this out and we agree that the sentence needed rephrasing. We changed "N cycling" to "$\delta^{15}$N" (line 26).

Line 24: 'δ15N' needs to be referred to consistently (soil δ15N, soil δ15N signature, stable isotope signature…are all used to refer to soil δ15N in the manuscript).

We revised the manuscript and tried to use 'soil $\delta^{15}$N' consistently.

Line 35: 'However' does not seem necessary

We have removed "However" (line 39).

Line 44: replace 'forest' by 'forests'

We replaced it accordingly (line 48).

Line 52: Delete 'activity'

We deleted the word "activity" (line 58).

Line 55: Correct 'intact' as 'an intact'

We corrected this accordingly (line 58).

Line 58-62: revise these sentences. Consider this 'Some studies from geometrically active sites of the tropics (Costa Rica and Taiwan) found lower N availability and more closed N cycle in steeper sloping positions suggesting that erosion has a significant control on N cycling (Hilton et al., 2013; Weintraub et al., 2015). However, and the magnitude of this effect in more stable landscapes is unknown calling for a consistent study across geomorphic gradients in the tropics.

We revised this sentence accordingly (lines 63 – 68).

Line 64 : Edits 'The stable isotope composition of N ($\delta$15N)' as 'The natural abundance of stable 15N isotope ($\delta$15N) of plant and soil pools'

This has been edited (line 70).

Line 66: should be 'insights into'

This has been changed (line 72).

Line 75-82: A testable hypothesis about the pattern soil $\delta$15N and N availability and openness of N cycle is needed. I would also question the hypothesis that soil $\delta$15N would be lower on steeper slopes because the erosion on steeper slopes removes fresh organic matter input from plants, which would continuously keep $\delta$15N of surface soil low compared to the deeper surface.

We rephrased the hypothesis to: "We hypothesized that soil $\delta^{15}$N values are highest in lowland tropical forest, which would be an indication for a higher N availability and openness of N cycle." (lines 97 – 98). We further hypothesize that topsoil $\delta^{15}$N is more depleted in steeper slopes compared to topsoil $\delta^{15}$N at more flat positions and more enriched compared to deeper soil layers of the same profile (line 98 - 101).

Line 93: Are both forests used in this study?

The sampled lowland forest catchment (260 ha) consists of these two forest subtypes. As we had a randomly spatial sample coverage, it is most likely that soils from both sub-types have been sampled. However, we did not identify all tree species at each sampling location to determine if it is a monodominant or mixed forest. We added a sentence so that it is clearer to the readers (lines 114 – 115).

Line 130: 'Laboratory' is more formal

We changed the header to "Sample analysis" (Line 155).

Line 135: provide $\delta$15N of the atmospheric N2

As atmospheric $N_2$ is the international standard, it has per definition an $\delta$15N value of 0 ‰ as the delta values expressing the differences vs. air. We added it to the manuscript (line 163) and also the formula to calculate the $\delta$15N values (Eq. 1).

Line 146-147: Why only these two sites were chosen?

As we focused on the effect of topography and soil erosion on the soil $\delta^{15}$N signature, only literature data with reported soil slope values of the samples were considered. To the best of our knowledge

no other studies had a sampling strategy with within site variation of slope angles. We added this explanation to the manuscript (lines 176 – 179).

Line 155 (last sentence): Consider putting it at the end of the paragraph

We put this sentence at the end of the paragraph (line 192 - 193).

Line 157-159: The SEM analysis was very simple with only five sites with only few potential factors that affect soil δ15N being included in the model. What are the variables included in the model?

As described above we only included soil $\delta^{15}N$ data, where slope angles of the samples were available, thus only 5 sites were included in the SEM. We included MAP, MAT, LAI, slope and soil C content as predictive variables for soil $\delta^{15}N$ in the model. Now we added this information to the manuscript (line 190 - 191).

Line 161-165: The values of these variables needs to be directly presented; it is not enthusiastic to many readers to extract the information from the Table (estimates).

The values for N stocks and C:N ratios are already mentioned in the text for each site. We also added the values for the $\delta^{15}N$ to the text, now it is easier for the reader to extract this information (lines 200 – 201).

Line 187: I would not use 'N cycling'. This study did not investigate the many aspects of N cycling. More importantly, the many factors known to affect soil δ15N and which are very important to interpret soil δ15N are not measured.

We agree that the title might be misleading, and we changed it to: "Using soil $\delta^{15}N$ to assess differences in ecosystem N-turnover" (line 222)

Line 188-89: Eshetu et al., 2004 Forest Ecology and Management 187, 139–
147 (Ethiopia) and Gerschlueret et al., 2019 Biogeosciences 16, 409–424 (Tanzania)
are some of the relevant references missing.

We thank the reviewer for providing additional references for our manuscript. So far, we listed only references from old growth natural tropical forests. We added the values from the semi-natural montane forest in Tanzania (Gerschlauer et al., 2019, line 228) and from an old-growth montane forest in Ethiopia (Eshetu and Högberg, 2000, line 228). Furthermore, we found an additional paper on soil $\delta^{15}N$ in a Miombo woodland in Tanzania (Mayes et al., 2019, line 240)

Line 207-208: This is not necessarily true as lower soil/plant δ15N is not always associated with limited N availability (closed N cycle). Gurmesa et al., 2017 Biogeosciences, 14, 2359–2370 (many other studies in SE Asia) have reported ecosystems pools can be strongly 15-depleted under N saturated condition.

We agree with the reviewer that the lower soil $\delta^{15}N$ is not always indicative of a more closed system as it is governed by a multitude of factors of the N cycle. For instance, in the Miombo, $N_2$-fixation is likely depleting the overall isotope signature of the system, which needs to be kept in mind when comparing these different forest types. Therefore, we added a sentence to this statement (lines 246-248) and subsequently discuss the influence of $N_2$-fixation on the soil $\delta^{15}N$ of the Miombo woodland (lines 248-252).

Line 209: how about the effect of δ15N of deposition N? Craine et al., 2015b?

We agree with the reviewer that N deposition influences the isotopic signature of soil N and we included this with the provided reference (line 253).

Line 214: depleted N-input from where? Only biological N deposition? Do you have data for N2-fixing plant species as well as their mycorrhizal association in the three forests? These are very crucial to interpret soil δ15N values.

While for the Miombo forest the depleted N-input is probably mainly from more $N_2$-fixing, the montane forest is more likely to receive depleted biological N input via deposition (Bauters et al., 2017). Unfortunately, we do not have data on $N_2$-fixing species available for our sites, but it is well documented this process is more important in the subtropical woodlands, compared to the tropical forests (Hogberg & Alexander 1995).

Line 236: this sentence does not help with the logical flow points being discussed in the paragraph

We thank the reviewer for pointing this out and agree that this sentence is indeed out of place. We removed this sentence from the revised manuscript (lines 299 - 300).

Lines 237-238: Line 226-227 repeated? Again, as I mentioned above, low soil δ15N does not necessary indicate closed N cycle. The context needs to be discussed. To say whether N cycle is dominated by organic N, it needs additional measurement. Is there data for soil inorganic N concentration in each forest?

We agree that the whole paragraph contained too many repetitions. We restructured the whole paragraph to have a better flow for the readers. We measured aquatic N exports for all the catchments and the montane forests exports slightly more dissolved organic N (67% of TDN is DON) than the lowland forest (61%). We added this data to a new Table 3.

Line 239: Edit 'excess of available N' as 'excess N availability'. However, it is not correct to conclude that the forests have excess N availability only based on the values of soil δ15N.

We moved the sentence up and changed it accordingly (line 284). Additionally, we added a statement that high N depositions and high $N_2$-fixation are an additional indicator for high N availabilities in these forests (line 285).

Line 240: It is amazing that the author did not provide data on N deposition for any of the sites (including those from literatures).

We present now N deposition data from the literature for all sites (Table 3).

Line 248: change 'soil N' to 'soil δ15N'.

This has been changed (line 316).

the discussion about effects of topography on soil δ15N is interesting, but it did not establish mechanistic relationship of topography with other factors known to strongly affect soil δ15N. The implication in discussion here is that soil δ15N is strongly affected by physical process (erosion) and the factors that control the erosion.

It is true that other factors than erosion influence soil $δ^{15}N$ (temperature, precipitation and vegetation cover). However, these factors did not vary within our sites and only the physical processes were potentially influenced by slope gradients. We added a sentence to the beginning of the paragraph to state our assumption (lines 317 – 319).

Line 289: 'samples' or 'sites?

We thank the reviewer for the attention to the detail. Sites is correct and this has been changed (line 360).

Few technical corrections /writing
Line 19: delete one of the 'in's

We deleted one of the 'in's.

Line 65: Should be Craine et al., 2015a. Also check line 209.

The citations have been amended (line 71 & 253)

Figure 2: first letter in y-axis label should be capitalized Figure 3: first letter in x-axis label should capitalized

The axis labelling of the two figures has been changed.

Table 2: Is it important to have all those decimals for fixed effect Estimates?

We rounded all values to have only 2 decimals for the fixed effects.

References Clarke et al., 2013 (Line 32) and Vitousek 1985 (line 40) are missing. The superscript in 15N or δ15N are not correctly written for many reference

At line 32 we did cite 'Alvarez-Clare et al., 2013' and not 'Clarke et al., 2013'. We added the reference of Vitousek 1984 and corrected the superscript for several references.

**Response to the interactive comment by the anonymous referee # 2**

The manuscript quantifies the nitrogen (N) stocks and N isotopic composition of soils at three locations in the Congo Basin. The aim was to explore N availability in ecosystems across this poorly studied region, in the broader context of understanding N cycling in tropical forests. As a key macronutrient, the N cycle of these forests is a critical part of understanding how an ecosystem might respond to external drivers (changes in pCO2, climate, landuse). The study finds large contrasts in the stable N isotopic composition (d15N) between the sites, alongside changes in N stock, and seeks to link these to differences in environmental and geomorphic variables. At each site, the work explores how slope angle (and topographic position) influence d15N, building on some past work in Taiwan and Costa Rica, to explore how geomorphic processes influence N cycling. The study was well focused, succinct, and the theme makes it worthy of attention at SOIL. However, I found the discussion quite hard to follow, and it was hard to draw out the main findings. My main comments below reflect this, and make some suggestions for revisions:

We thank the reviewer for the insightful comments and constructive review. We address all the reviewer's comments below and we feel that - through a revision of the manuscript- the MS quality has greatly improved.

1) Provide a clearer assessment of the potential controls on d15N in soil: This doesn't have to be more than a paragraph, as this has been done in other papers (from time to time), but the paper lacks a clear explanation of what controls the d15N values of soil N. This would be useful in the introduction, and then used to seed the structure of the discussion and help a clearer assessment of

what best explains the patterns in the data. I would suggest something that talks about N inputs (and their d15N values), internal N cycling (plant to soil) and role of N losses (gaseous, dissolved, particulate) and how they may fractionate (or not) N isotopes in soil. Some of this is there in the manuscript, but its not that clear, and confused by the "open" vs "closed" discussion (see next point).

We thank the reviewer for raising this point. We agree that the description of the factors controlling soil $\delta^{15}N$ signatures would be well placed in the intro. We revised the fourth paragraph in the introduction and added more information on the different processes controlling soil $\delta^{15}N$ (lines 73 – 82).

2) The "open" vs "closed" explanation for d15N values: This seems too simplified now, as we recognise that we can vary several aspects of the N cycle in an ecosystem and arrive at the same d15N values. For instance: i) the comparison between the N stock (N/km2) and input and output fluxes (N/km2/yr) can play a role, as with any isotope mass balance; ii) the N inputs (deposition, fixation) can be fractionated (or not); iii) the N outputs (gaseous, dissolved, particulate) can be fractionated (or not); iv) and pedogenesis and timescales of soil formation can vary (giving different intergration periods for different sites, and over depth). So with this explanation at hand, the simple argument of closed vs open is simplistic. In fact, the open vs closed model (I think) implicitly assumes that all N losses are fractionating, and that the ratio of N stock to N fluxes are the same at every site. Both those assumptions are flawed. Instead, this study measures N stocks (and C/N, so relative to C). So it can say something about how this varies (and the paper doesn't use this information paired to the d15N data).

The 'open' vs 'closed' system approach is widely used in literature. It is one way to interpret the scarce data available. However, we agree that the explanation is far from perfect and all the points mentioned by the reviewer will also affect the soil $\delta^{15}N$ values, and hence need to be addressed accordingly in the discussion. We see that we need to improve the way of discussing our data and put more emphasis on all the possible processes influencing measured soil $\delta^{15}N$ values. We restructured the whole paragraph 4.1 (see point 3 of reviewer) and more processes that can alter soil $\delta^{15}N$ are being discussed.

The study doesn't measure plant d15N, NO3 in porewaters or streams, or any gaseous N (that is very rare to do). This means any discussion of these important features of the N cycle and their d15N values would have to be drawn from other studies, and somewhat speculative for these sites. However, at the moment the paper doesn't discuss at all what these could be, and whether they could vary between the sites. By way of example, the lower MAP at the Miombo site could influence soil moisture – which is important for gaseous N loss (under saturated conditions) and NO3 loss (which can have a low d15N value). Thus, this could explain the shift in isotopic values: this site has less fractionating N losses. Or could it be simply a plant input (fixation) story. Another quick example, the montane and lowland sites have similar d15N values, but the lowland site has much lower N stock (but higher relative to carbon C/N). So, to get the same d15N depletion in the soil residue, one has to invoke that the N fluxes out of the system (which fractionate) are larger in the montane system, than the lowland (because to see a d15N shift, you need the flux to be larger). This text from me is somewhat off the top of my head. I could be completely off the mark here. But my point is that there are details to the dataset which are not discussed clearly, and the open vs closed discussion constrains this discussion in my view. A more structured discussion (see below) could also help.

We thank the reviewer for pointing out this unclarity, we highly appreciate this.
Indeed, we agree with the reviewer that a more detailed discussion about the different possible mechanisms which can influence $\delta^{15}N$ signatures is needed. The more depleted values in the Miombo woodland are most likely due to more $N_2$-fixation, occurring in this ecosystem, compared to the tropical forests. The stable isotopic signature of soil N in the montane forest is especially in topsoil a lot lower compared to the signature in the lowland forest, which is indicating that N inputs

are depleted and/or less fractionating processes (or more erosion) are happening in the topsoil. However, the different shape of the $\delta^{15}$N profiles tend to indicate different N availabilities as the shape present in the montane forest is typical for an N-limited ecosystem (increasing values with depth) while the shape in the lowland forest is more typical in N-rich ecosystems (highest $\delta^{15}$N value in an intermediate depth). We included now the gaseous N and leaching losses for all the sites (Table 3) and discussed them accordingly (lines 276-280).

3) Discussion section: I would recommend restructuring this to either take a more site by site explanation of patterns. Or a process by process explanation of patterns (e.g. starting with potential N inputs – could these explain things; then differences in N stocks; then potential N outputs). This could help draw out the key take away messages a little better.

To improve the structure, we are glad to apply the suggestion by the reviewer and restructure the section 4.1. We now started the paragraph by discussing topsoil $\delta^{15}$N values for all three sites and the controls resulting in these signatures (N$_2$-fixation, litterfall input and N deposition). Then we proceed to discuss the differences in the depth profiles of three sites and the processes responsible for it (denitrification, leaching).

Note – only having completed my review did I then read the comments already posted in the discussion. I found myself in agreement with queries flagged by the other reviewer.

We are glad that both reviewers are agreeing on the points mentioned. We addressed everything from reviewer 1 above.

Other comments (with line number):
19: maybe avoid the word "profiles" here – as the reader could infer you're talking about a soil profile, with depth.

We indeed mean the soil $\delta^{15}$N distribution over the depth of the soil profile.

19-20: this sentence would be better linked to the variability in d15N values measured, and how they've been interpreted.

We moved the sentence up so that confusion is avoided (lines 17 – 19).

23: this sentence on montane forest was a little confusing following the preceeding sentences, and perhaps the order of information here needs to be revised.

To make it more coherent we changed this sentence to: "Despite the steep topography, slope angles do not constrain soil $\delta^{15}$N in the montane forest, although this ecosystem experiences high variability in the stable isotope signature." (lines 26 – 27).

44: can the sentence "it is important" be rephrased to better spell out what the knowledge gaps are?

With this sentence we wanted to emphasize that different tropical forests are highly variable in nutrient availabilities and that a generalization of tropical forests being rich in N and poor in P is not suitable. This statement summarizes the points made before in the paragraph and helps to reason our research.

46: the "openness" section of text. I wonder if you need a couple of sentences explaining the inputs of N to ecosystems, and the losses. And then the idea that the overall size of the pool and leakiness is conceptualised as open vs closed. This might be clearer to those not familiar with the N cycle in soils.

We added some information on in- and outputs of N into the system (lines 53-54).

60: I partly agree with that statement... But there is an important detail - Hilton et al., don't invoke the open vs closed concept. Instead, they argue that the nature of the N loss varies with slope, and that physical erosion and export of organic N in solid form does not fractionate the N isotope pool. In that way, the isotope mass balance is different for sites on steeper slopes (N loss dominated by non-fractionating losses), vs shallower slopes which potentially have a greater role of fractionating N losses (dissolved N forms, N gas forms).

We agree that the concept of open vs. closed systems might not fit best for the studies cited here. We changed the sentence in line 65. The effect of physical soil loss on the $\delta^{15}$N signature is described later in the introduction.

69: please expand on the "openness of the N cycle" comment.

We extended this statement and hope it is now easier to understand for the readers (lines 84-86)

70: yes this is exactly what I write above! I should have been patient. Anyhow, I think perhaps that means that the order and flow of content might need some edits here.

In the paragraph before we explained differences in N availability on regional and local scale and in this paragraph, we discuss the influence of different processes on the soil $\delta^{15}$N. We hope that with the new structure it is easier to follow.

105: experimental design seems good – and impressive range of sites across this setting. A quick Q – do you know the bedrock geology and whether it varies (and whether it could contain N?).

We thank the reviewer for acknowledging our experimental design and we extracted information on the parent material of our study sites from the soil and terrain database for Central Africa (SOTERCAF) and included it in the site description (lines 108, 116 & 124).

Figure 1 – please add a note to the caption that the colours are elevation (I guess?) and perhaps make a note of the resolution of the DEMs shown here.

Indeed, the colors show different evaluations of the sampling sites. We added this information and the DEM source to the figure caption.

Table 1 – is there a typo here? The lowland forest has the highest mean slope (22degrees) – which doesn't seem to fit with what you have shown in the histograms of slopes in Figure 1.

The observation of the reviewer is right, the values for the mean slopes of the lowland and montane sites were swapped. This has been changed in the new version of the manuscript. We thank the reviewer for pointing this out.

135: briefly detail the external standards used to re-calibrate the d15N values and their precision etc.,

This information has been added to the manuscript (line 159).

140: adapted or used?

We used the model calculation of Pelletier 2012 and we changed the text accordingly (line 169).

138: a bit more context on why this model was selected would be useful.

We assume that the reviewer wants more context on why the SEM model was selected. We used a structural equation model as possible dependencies between variables can be included. We added more information to the manuscript (lines 190 – 191).

Figure 3 B – how did you lump the sites together to get this erosion coefficient?

As described in equation 1, the EC is calculated from the slope, MAP and the LAI. To calculate average EC values for each site, the average slope of every single sample from each site was used to calculate the respective EC.

Section 4.1. – I found this hard to follow. There is some repetition of themes and information (especially in the final paragraph), and it was hard to take away the main discussion points the authors wanted to highlight. It might make sense to start with a discussion of the N inputs, and the top soil values (and their contrasts) and what that indicates about them. The discussion N outputs/internal cycling (and depth profiles) at each site. And try to draw together a somewhat coherent discussion. One of the striking things is how high the d15N values are in the lowland (and at depth in the montane) and I finished this section without a clear idea what that was being attributed to.

As already indicated above, we restructured the discussion in section 4.1. The higher $\delta^{15}$N values in the lowland topsoil is most likely to less depleted N inputs compared to the Miombo woodland (less $N_2$-fixation). The isotopic values of the $\delta^{15}$N profile of the montane forest are steadily increasing with depth and this is a typical shape of profile of an N limited ecosystem (Hobbie and Ouimette, 2009). We rephrased the whole paragraph and hope now that the underlying processes are clearer to the reader.

Table 2: I don't understand the "Estimate" values in this table, and struggle to follow what they refer too.

The values in table 2 show the results for all the fixed effects estimated by the linear-mixed effects model (Estimate) and the respective standard errors and P-values. A presentation like this lets the reader easily find the mean values for each response variable and it is clearly visible which effects differ significantly from others. We changed the table from "Estimate" to "Effect size" to make it clearer for the readers

251 – "there are no steep slopes in the lowland forest" – this does suggest that Table 1 is incorrect.

As mentioned above the slope values in Table 1 are switched between montane and lowland forest and we changed it in the revised manuscript. That was an important typo to find. Thanks again!

255: more about the controls on the EC output would be useful – as to why Miombo is so much higher. And how you computed the EC values for the literature data. And how Figure 3B came about (and the assumptions and limitations associated with it).

The driving variable behind the high EC value in the Miombo forest is mainly the low LAI, as we already stated in the text. The forest cover in the Miombo is less dense compare to the other forest ecosystems and thus the soil is less protected from the erosive force of rainfall events. In section 2.4 we described how we calculated the EC values and where we obtained the data for the values of the literature.

301: this note on N fixation was not clearly discussed in the main text – see comment

above on Section 4.1

We discussed the possible influence of fixed $N_2$ on the $\delta^{15}N$ signature in the Miombo woodland in section 4.1 L 245-250. We revised the whole section 4.1 in the new manuscript and hope that also the $N_2$-fixation part is now better discussed.